# Neuronal CaMKK2 promotes immunosuppression and checkpoint blockade resistance in glioblastoma

William H. Tomaszewski [1] ✉, Jessica Waibl-Polania[2], Molly Chakraborty [3], Jonathan Perera [3], Jeremy Ratiu [1], Alexandra Miggelbrink [2], Donald P. McDonnell [4], Mustafa Khasraw [5,6], David M. Ashley[5,6], Peter E. Fecci [1,2,5,6], Luigi Racioppi [7,8], Luis Sanchez-Perez[5,6], Michael D. Gunn [1,2,9] & John H. Sampson [1,2,3,5,6]

Glioblastoma (GBM) is notorious for its immunosuppressive tumor microenvironment (TME) and is refractory to immune checkpoint blockade (ICB). Here, we identify calmodulin-dependent kinase kinase 2 (CaMKK2) as a driver of ICB resistance. *CaMKK2* is highly expressed in pro-tumor cells and is associated with worsened survival in patients with GBM. Host CaMKK2, specifically, reduces survival and promotes ICB resistance. Multimodal profiling of the TME reveals that CaMKK2 is associated with several ICB resistance-associated immune phenotypes. CaMKK2 promotes exhaustion in CD8⁺ T cells and reduces the expansion of effector CD4⁺ T cells, additionally limiting their tumor penetrance. CaMKK2 also maintains myeloid cells in a disease-associated microglia-like phenotype. Lastly, neuronal CaMKK2 is required for maintaining the ICB resistance-associated myeloid phenotype, is deleterious to survival, and promotes ICB resistance. Our findings reveal CaMKK2 as a contributor to ICB resistance and identify neurons as a driver of immunotherapeutic resistance in GBM.

Immune checkpoint blockade (ICB) has revolutionized treatment for many difficult-to-treat cancers but has yet to produce significant improvement in outcomes for patients with glioblastoma (GBM)[1,2]. The etiology of ICB resistance in GBM remains to be fully elucidated but is thought to be linked to the immunosuppressive nature of the tumor microenvironment (TME) and the pro-tumor function of stromal cells, including tumor-associated macrophages (TAMs)[3], also sometimes referred to as glioma-associated macrophages in GBM. Likewise, the GBM TME is particularly immunologically "cold," with relatively poor T-cell infiltration[4,5]. Therapeutically targeting pro-tumor stromal cells

is a promising avenue for converting a TME from "cold" to "hot", while improving responses to immunotherapy.

TMEs have been characterized in a variety of cancers using transcriptomic analysis to more accurately predict ICB responsiveness. For instance, TMEs enriched for antigen presentation (*HLA-A, TAP1, CIITA, HLA-DRA*), cytotoxicity (*GZMB, GZMA*), T-cell trafficking (*CXCL9, CXCL10*), and Th1 (*CD4, IFNG, CD40L*) immune signatures are more likely to respond to ICB therapy[6]. Certain cell types within the TME have also been found to be strongly associated with ICB response or resistance. For instance, *TREM2*⁺, *SPP1*⁺, *APOE*⁺ TAMs were recently

[1]Department of Immunology, Duke University Medical Center, Durham, NC, USA. [2]Department of Pathology, Duke University Medical Center, Durham, NC, USA. [3]Department of Biomedical Engineering, Duke University, Durham, NC, USA. [4]Department of Pharmacology and Cancer Biology, Duke University Medical Center, Durham, NC, USA. [5]Preston Robert Tisch Brain Tumor Center, Duke University Medical Center, Durham, NC, USA. [6]Department of Neurosurgery, Duke University Medical Center, Durham, NC, USA. [7]Department of Medicine, Division of Hematological Malignancies and Cellular Therapy, Duke University School of Medicine, Durham, NC, USA. [8]Department of Molecular Medicine and Medical Biotechnology, University of Naples Federico II, Naples, Italy. [9]Division of Cardiology, Department of Medicine, Duke University Medical Center, Durham, NC, USA. ✉e-mail: william.tomaszewski@duke.edu

identified as an immunosuppressive, pro-tumor population conserved across multiple human tumor types[7], including GBM[8]. Genetic or therapeutic inhibition of Trem2, a hallmark gene of the disease-associated microglia (DAM) phenotype, was recently shown to license ICB therapy in murine tumor models[9,10]. This indicates that TAMs sharing characteristics with DAMs are conserved throughout human cancers—including GBM—and play a role in fostering ICB resistance and immunosuppression in the TME. In addition to TAMs, tumor-infiltrating lymphocyte (TIL) phenotypes, particularly as they reflect exhaustion, also strongly predict ICB response[11,12]. In mice and humans, precursor-exhausted CD8+ TILs (Tcf1+, Slamf6+, Pd1+) remain responsive to ICB, whereas terminally exhausted CD8+ TILs (Tim3+, Tox+, Pd1+) do not[13–15]. Limiting the pro-tumor functions of TAMs in the GBM TME and promoting functional TIL phenotypes associated with an ICB response are strategies anticipated to newly license immunotherapies against GBM.

Recent studies have demonstrated that many pro-tumor processes can be attributed to calcium signaling in stromal cells in response to tumor stimuli[16], including in GBM[17]. Calmodulin-Dependent Kinase Kinase 2 (CaMKK2) is a calcium-responsive serine-threonine kinase[18,19]. Elevated CaMKK2 activity is found within prostate[20], hepatic[21], and breast[22] cancers, while loss of CaMKK2 has been shown to polarize TAMs to an anti-tumor phenotype in murine breast cancer and lymphoma models[22,23]. CaMKK2 is highly expressed in macrophages and neurons[18,24], both of which are abundant stromal cells in GBM and possess pro-tumor function[3]. Accordingly, CaMKK2 is critical for brain-derived neurotrophic factor (BDNF) expression in neurons[25,26], which has already been seen to contribute to pro-tumor mitogenic functions in high-grade gliomas[27,28]. Indeed, it has previously been shown that elevated CaMKK2 expression is associated with a worse prognosis in patients with GBM using The Cancer Genome Atlas (TCGA) and the Chinese Glioma Genome Atlas (CGGA)[29]. Collectively this suggests that CaMKK2 represents a potential therapeutic target that promotes pro-tumor functions of stromal cells and is highly expressed in the GBM TME. We hypothesize that the expression of CaMKK2 is critical for the pro-tumorigenic GBM TME and ICB resistance.

In this work we demonstrate that neurons contribute to ICB resistance, that CaMKK2 is highly expressed in neurons and TAMs and has tumor-promoting functions in the GBM TME.

## Results

### CaMKK2 expression within the GBM TME is associated with poor survival and resistance to ICB

Using CaMKK2-EGFP reporter mice, we confirmed that CaMKK2 is most highly expressed in TAMs and neurons in naïve and tumor-bearing mice (Fig. 1a and Supplementary Fig. 1a–c). Publicly available scRNA-seq data demonstrated that this pattern of expression was similar in the naïve human brain and patient GBM samples (Fig. 1b and Supplementary Fig. 1d). Survival analysis of data from the Gene Expression database of Normal and Tumor tissues 2 (GENT2) database[30] subsequently showed that high levels of CaMKK2 expression within the combined tumor and TME are associated with worse survival outcomes in patients with GBM (Fig. 1c).

To resolve whether CaMKK2 in stromal cells or the tumor itself might drive tumor progression, we sought to determine if CaMKK2 deficiency in the tumor-bearing host was sufficient to extend survival in syngeneic orthotopically implanted glioma models. Indeed, CaMKK2−/− (CaMKK2 KO) mice survived significantly longer than wildtype (WT) mice implanted with three separate GBM tumor models: CT2a, GL261, and KR158B-Luc (Kluc) (Fig. 1d–f).

To reveal the difficulties imposed upon ICB treatment specifically by the intracranial (IC) TME, we compared the efficacy of a combination therapy consisting of ICB antibodies against programmed cell death protein 1 (PD1) and T-cell immunoglobulin and mucin-domain

containing-3 (TIM3) in subcutaneous (SQ) and IC GBM models in WT mice. This combination ICB therapy was selected based on the high expression of PD1 and TIM3 on TILs isolated from patients with GBM[31]. While ICB combination therapy was efficacious in inducing tumor regression in WT mice in the SQ model, it had no significant effect on the survival of WT mice in the IC model (Supplementary Fig. 1f and Fig. 1g, i). These findings suggest that factors present in the WT IC TME but absent in the SQ TME contribute to ICB resistance. To determine whether CaMKK2 might contribute to ICB resistance within the IC compartment, we compared treatment efficacy against IC CT2a in WT and CaMKK2 KO mice. CT2a was chosen for this and other experiments due to its low immunogenicity[32], and histological similarities with human GBM[33]. ICB treatment produced a significant survival benefit in the CaMKK2 KO mice (Fig. 1g). This finding demonstrates that CaMKK2 contributes to ICB resistance in IC GBM.

Considering the newly conferred response to ICB, a traditionally CD8+ T-cell-directed therapy, we evaluated whether T cells mediate tumor clearance in CaMKK2 KO mice. Utilizing anti-CD8 depleting antibodies, we confirmed that the survival benefit in CaMKK2 KO mice is immune-mediated and dependent on CD8+ T cells (Fig.1h, j). Broad immunophenotyping of the TME showed that the number of tumor-infiltrating lymphoid and myeloid cells, however, was not significantly altered by CaMKK2 deficiency (Supplementary Fig. 1g), suggesting it may instead be phenotypic or localization differences in the immune compartment that promote survival in the absence of CaMMK2.

### Pro-tumor immune programming in the glioma microenvironment is CaMKK2 dependent

As the number of tumor-infiltrating lymphoid and myeloid cells were not significantly altered in CaMKK2 KO GBM-bearing mice, we investigated the contribution of CaMMK2 to tumor-infiltrating immune cell phenotypes. We performed scRNA-seq on CD45+ immune cells isolated from IC CT2A tumors in WT and CaMKKK2 KO mice (Fig. 2a, b). Cell clusters were annotated using previously published gene expression signatures (Fig. 2d and Supplementary Fig. 2a).

Indeed, due to the higher dimensionality of scRNA-seq data relative to flow cytometry data, phenotypic differences among the major immune subsets became apparent. Heterogeneity was observed within the TAM compartment, as three clusters were identified that highly expressed canonical macrophage genes (Mertk, Adgre1, Fcgr1). These three TAM clusters were labeled Nos2+ TAM (Nos2 and Arg1), DC-like TAM (H2-Aa and CD74), and Apoe+ TAM (Apoe and Mrc1) (Fig. 2d). Apoe + TAMs were nearly exclusive to the WT TME, and DC-like TAMs were predominantly found in CaMKK2 KO mice (Fig. 2b, c). The dramatic shift in these immune populations suggests a role for CaMKK2 in maintaining the Apoe+ TAM phenotype and in restricting antigen presentation phenotypes amongst TAMs.

### CaMKK2 deficiency averts a terminally exhausted phenotype in tumor-infiltrating CD8+ T cells

Given the increase in immunostimulatory programming of the mononuclear phagocyte (MNP) system (bone marrow-derived TAMs, and Monocytes as well as yolk-sac derived tissue-resident Microglia), as well as the requirement for T cells in mediating the survival benefit in CaMKK2 KO mice, we further analyzed the TIL compartment. Reclustering of the CD3e+ TIL populations resolved additional populations (Fig. 3a), including Tregs (Ikzf2, Foxp3, Ctla4), effector CD4+ T cells (Cd4, CD40lg, IFNg, TNFa), CD8+ T cells (Cd8, Pdcd1, and Lgals3), stem-like T cells (Tcf7, Slamf6, S1pr1), and γδ T cells (Trdv4, Rorc, Sox13, Aqp3) (Fig. 3b).

Examining the requisite CD8+ T-cell population more specifically, we performed differential expression analysis on the reclustered CD8+ TIL population. This revealed that Granzyme B (Gzmb) and Granzyme A (Gzma) expression were both more highly expressed in CD8+ TILs found in CaMKK2 KO mouse tumors relative to those in WT (Fig. 3c).

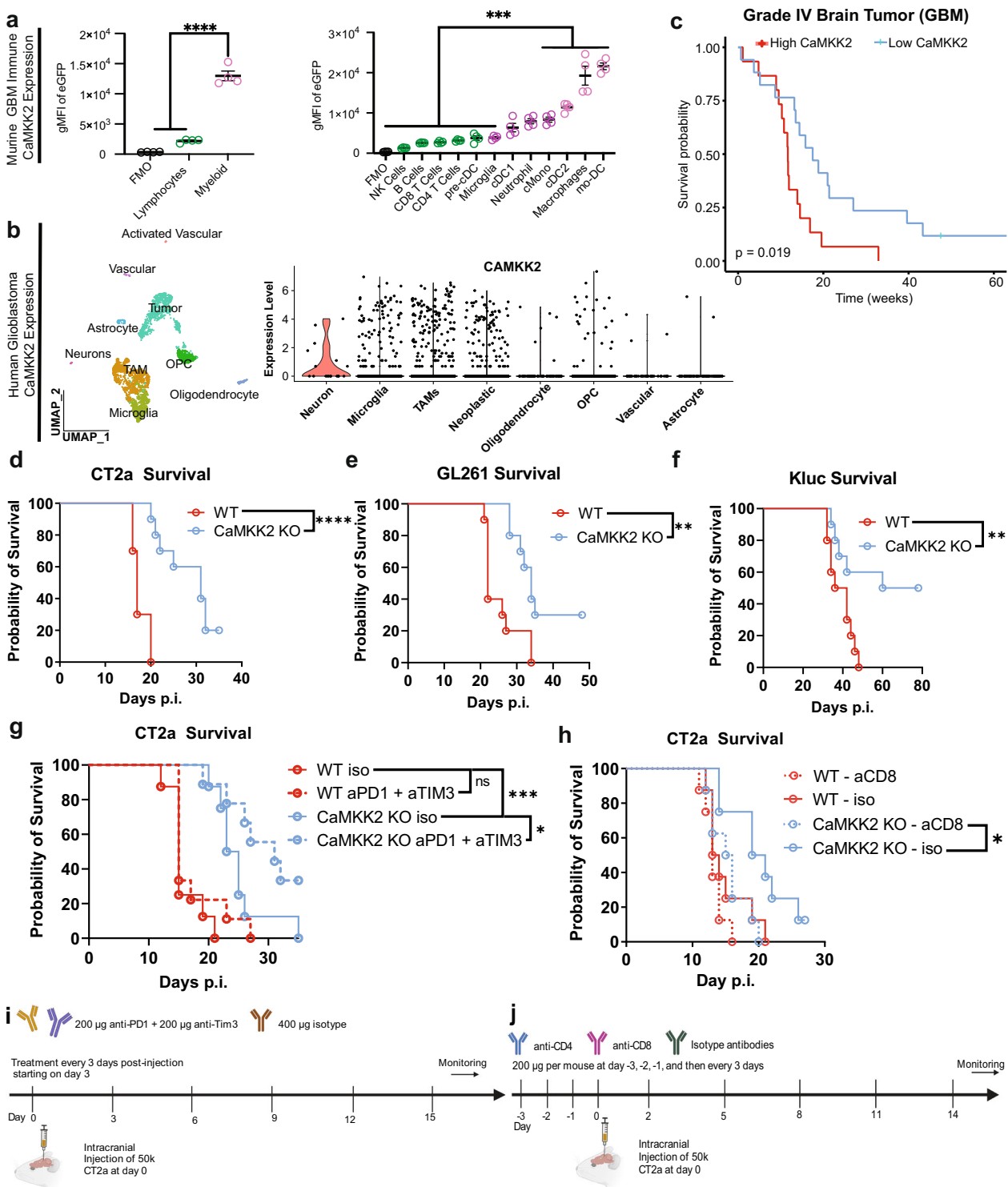

**Fig. 1 | CaMKK2 expression within the GBM TME is associated with poor survival and resistance to ICB. a** Tumor-bearing hemispheres were harvested from WT (FMO) or CaMKK2-EGFP mice on D14 and stained with a multi-color flow panel to resolve major immune populations. $n = 4$ mice per genotype, one-way ANOVA $p < 0.05$ with unadjusted post hoc two-tail Fisher LSD $t$-test. Data are presented as mean ± SEM. ****$p < 0.0001$, ***$p = 0.0004$. **b** UMAP plot of Human Glioblastoma tumor microenvironment, and Violin plot of CaMKK2 expression in corresponding cell types. Data pulled from http://gbmseq.org/ and re-analyzed. **c** Glioblastoma survival stratified by median Log2(CaMKK2) expression where above median expression was considered "High" and below "Low". High $n = 15$, low $n = 17$, Log-rank test. **d−f** Mice were intracranially implanted with 50k CT2a, 100k GL261, or 50k

Kluc and monitored for survival. $n = 10$ per group, Log-rank test. **d** ****$p < 0.0001$. **e** **$p = 0.0027$. **f** **$p = 0.0086$. **g** 50k CT2a was implanted intracranially and either 400 ug isotype or a combination therapy of 200 ug aPD1 and 200 ug aTIM3 was administered on D3 p.i and every 3 days through D15 for a total of 5 treatments. $n = 9$ for ICB treated groups and $n = 8$ for isotype treated groups, Log-rank test. ***$p = 0.0001$, *$p = 0.0416$. **h** 50k CT2a was implanted intracranially and mice were monitored for survival. Either 200 ug of isotype, aCD4 or aCD8 was administered on D-3,D-2,D-1, p.i. and every 3 days after through D14 for a total of 8 treatments. $n = 8$ per group, Log-rank test. *$p = 0.0279$. **i** Schematic depicting ICB treatment strategy. **j** Schematic depicting CD4 and CD8 depletion strategy. **i, j** Graphics were created with BioRender.com. Source data are provided as a Source Data file.

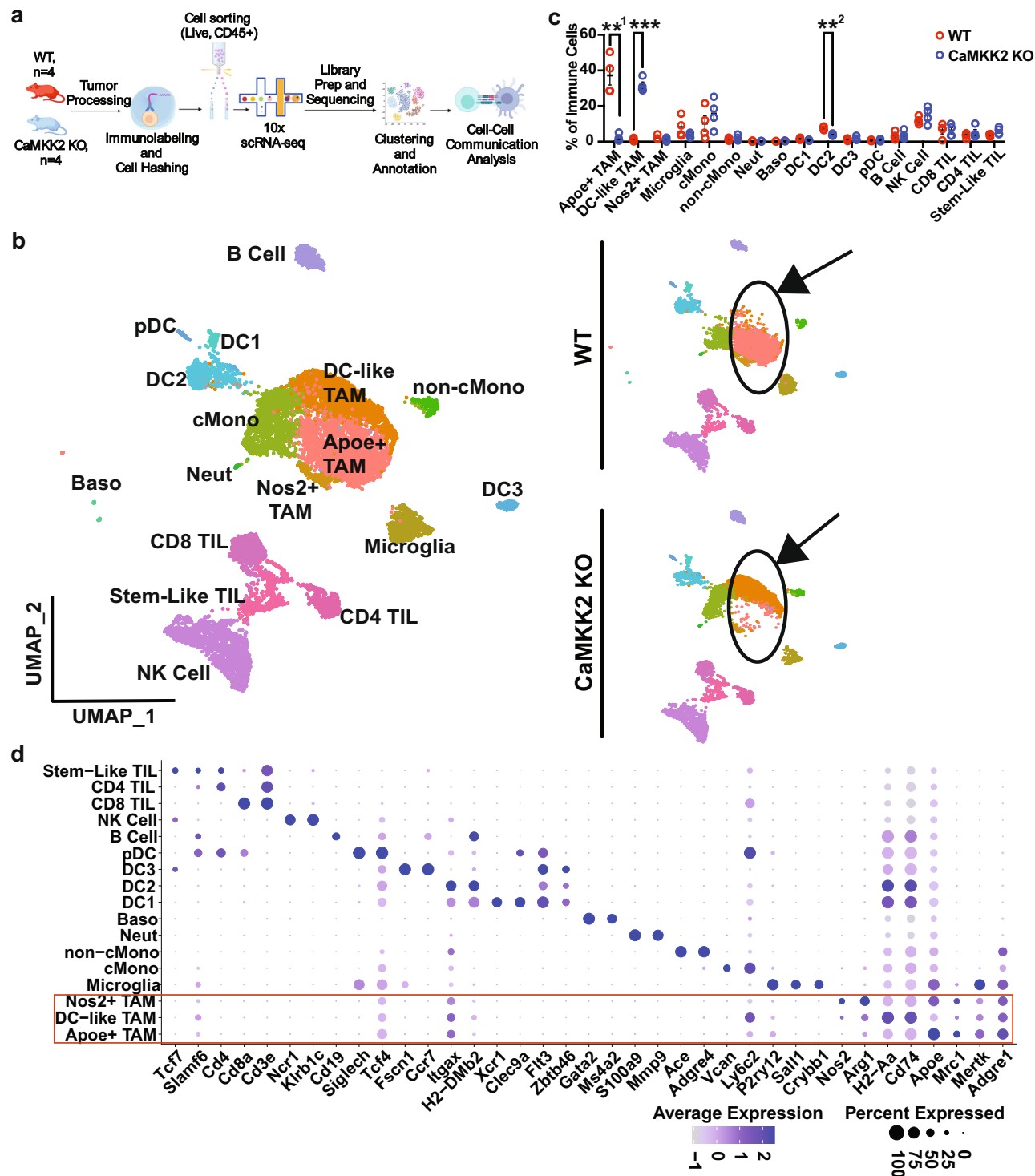

**Fig. 2 | Pro-tumor immune programming in the glioma microenvironment is CaMKK2 dependent. a** Schematic depicting scRNA-seq experimental design. $n = 4$ WT and CaMKK2 KO mice had 50k CT2a orthotopically implanted. On D14 tumors were harvested and underwent tumor processing. Samples were labeled with a Live-Dead viability Dye, CD45, and TotalSeqB anti-mouse Hashtag antibodies. CD45[+] Live cells were sorted, and then pooled in equal parts by genotype. Graphic was created with BioRender.com. **b** 14k CD45+ Live Immune cells were sorted from WT and CaMKK2 KO tumor-bearing hemispheres on D14 p.i. HTO and Gene Expression libraries were prepared using the 10X platform. UMAP plots of the cell types identified using unsupervised clustering methods are shown for the aggregate dataset and stratified by genotype. **c** Abundance of immune cell types identified by scRNA-seq. $n = 4$ per genotype, two-way RM ANOVA $p < 0.05$ with post hoc unadjusted two-tail Fisher LSD $t$-test. **[1]$p = 0.0059$, ***$p = 0.0003$, **[2]$p = 0.0016$. Data are presented as mean ± SEM. **d** Dot plots corresponding to the cell types displayed in the UMAP plots show expression of subset-specific genes, with the dot size representing the percentage of cells expressing the gene and the color representing its average expression within a cluster. Source data are provided as a Source Data file.

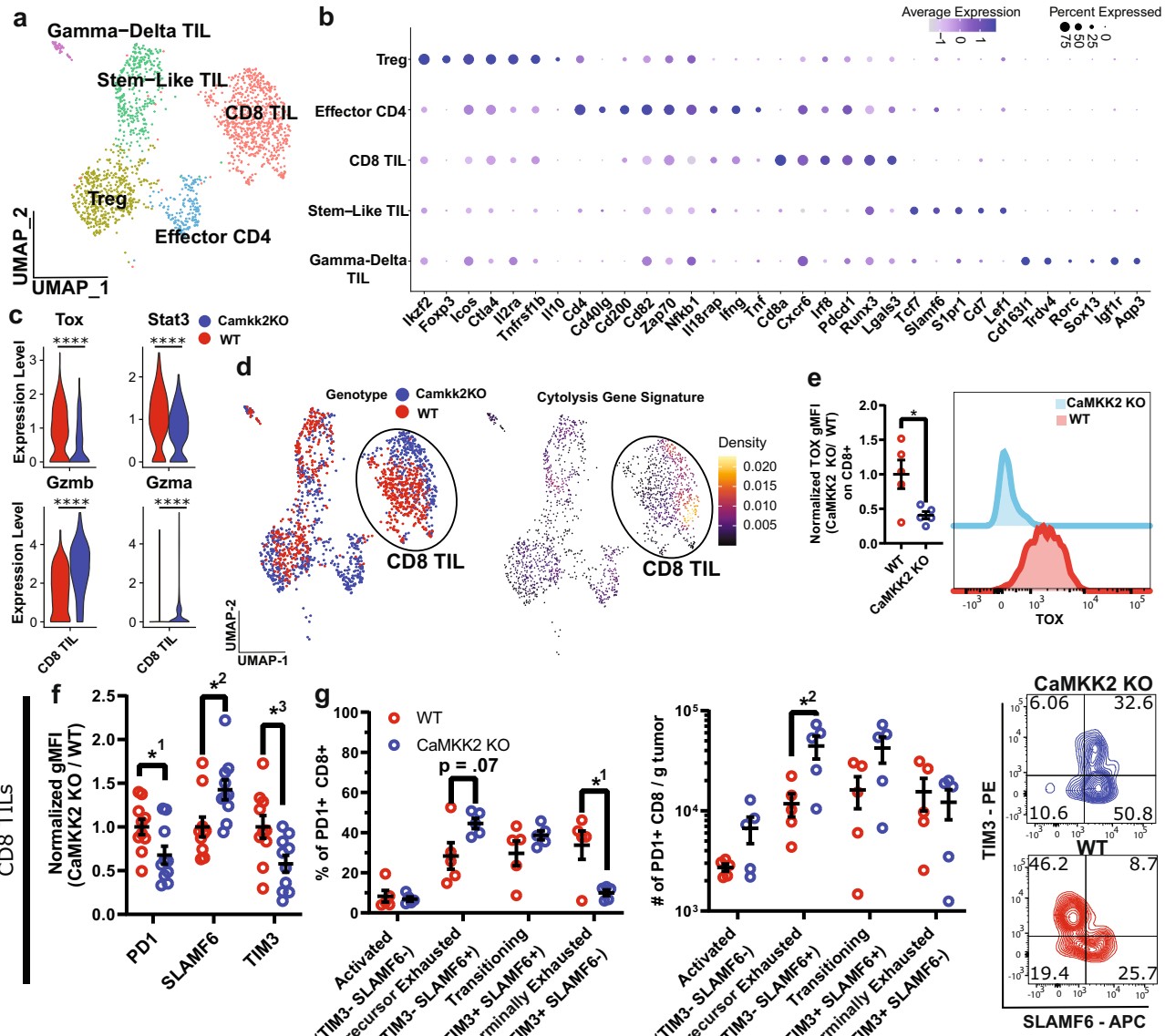

**Fig. 3 | CaMKK2 deficiency averts a terminally exhausted phenotype in tumor-infiltrating CD8⁺ T cells. a** UMAP plots of reclustered TILs presented in aggregate **b** Dot plots corresponding to the cell clusters displayed in the UMAP plot in panel **a** show expression of subset-specific genes. **c** Violin plots comparing exhaustion and cytolytic genes expression in CD8⁺ TILs. MAST was used for differential expression analysis. $n = 366$ WT and $n = 350$ CaMKK2. ****adj. $p < 0.0001$. **d** UMAP of reclustered TILs (same embedding as panel **a**) stratified by genotype, and a density plot of the cytolysis gene signature projected into UMAP space. **e** Tumor-bearing hemispheres were harvested on D14 post CT2a implantation and stained with a multi-color flow panel to assess Tox expression in CD8⁺ TILs. $n = 5$ per genotype,

two-tailed *t*-test. *$p = 0.0243$. Each sample replicate was normalized to the average WT gMFI. **f**, **g** Tumor-bearing hemispheres were harvested on D14 post CT2a implantation and stained with a multi-color flow panel to assess abundance and accumulation of terminally exhausted and precursor-exhausted T-cell phenotypes. **f** $n = 10$, two-way RM ANOVA $p < 0.05$ with post hoc unadjusted two-tailed Fisher LSD *t*-test. *¹$p = 0.03$, *²$p = 0.0167$, *³$p = 0.0191$. Results are combined from two independent experiments. Each experimental replicate was normalized to the average WT gMFI. **g** $n = 5$, two-way RM ANOVA $p < 0.05$ with post hoc unadjusted two-tail Fisher LSD *t*-test. *¹$p = 0.027$, *²$p = 0.0443$. **e**–**g** Data are presented as mean ± SEM. Source data are provided as a Source Data file.

Gzmb and Gzma are molecules well-known to be involved in cytolysis. Projection of the cytolytic gene signature onto the reclustered TILs shows that the signature strongly co-localized with the CaMKK2 KO CD8⁺ TILs (Fig. 3d). In contrast, the expression of *Tox* and *Stat3*, which regulate CD8⁺ T-cell exhaustion[34–37], were significantly lower (Fig. 3c). Intracellular flow cytometry staining of CD8⁺ TILs from CaMKK2 KO mice confirmed that Tox expression was also significantly lower at the protein level (Fig. 3e). This suggested that CD8⁺ T cells in the CaMKK2-deficient TME become less exhausted in the setting of GBM.

An ICB-responsive, precursor-exhausted CD8⁺ TIL population has been identified as Slamf6⁺, Tcf1⁺, Pd1⁺, and Tim3⁻[13–15]. The precursor-exhausted population shares characteristics with the stem-like TILs identified through reclustering (*Tcf7, Slamf6*)

(Fig. 3b). Because we observed a slight enrichment for stem-like TILs and reduced Tox expression in CD8⁺ TILs in CaMKK2 KO mice (Fig. 2c), we assayed exhaustion phenotype markers on CD8⁺ TILs by flow cytometry. We observed significantly reduced expression of PD1 and TIM3 (terminal exhaustion), and increased expression of SLAMF6 (precursor exhaustion) on CD8⁺ TILs in the tumors of CaMKK2 KO mice (Fig. 3f). Accordingly, within PD1⁺ CD8⁺ TILs, there was an increase in the prevalence of the precursor-exhausted phenotype (SLAMF6⁺, TIM3⁻), and reduced occurrence of the terminally exhausted phenotype (SLAMF6⁻, TIM3⁺) (Fig. 3g). The increased proportion of precursor-exhausted CD8⁺ TIL phenotypes was reflected in an increase in absolute counts of precursor-exhausted CD8 T cells per gram of tumor (Fig. 3g). This suggested that CaMKK2

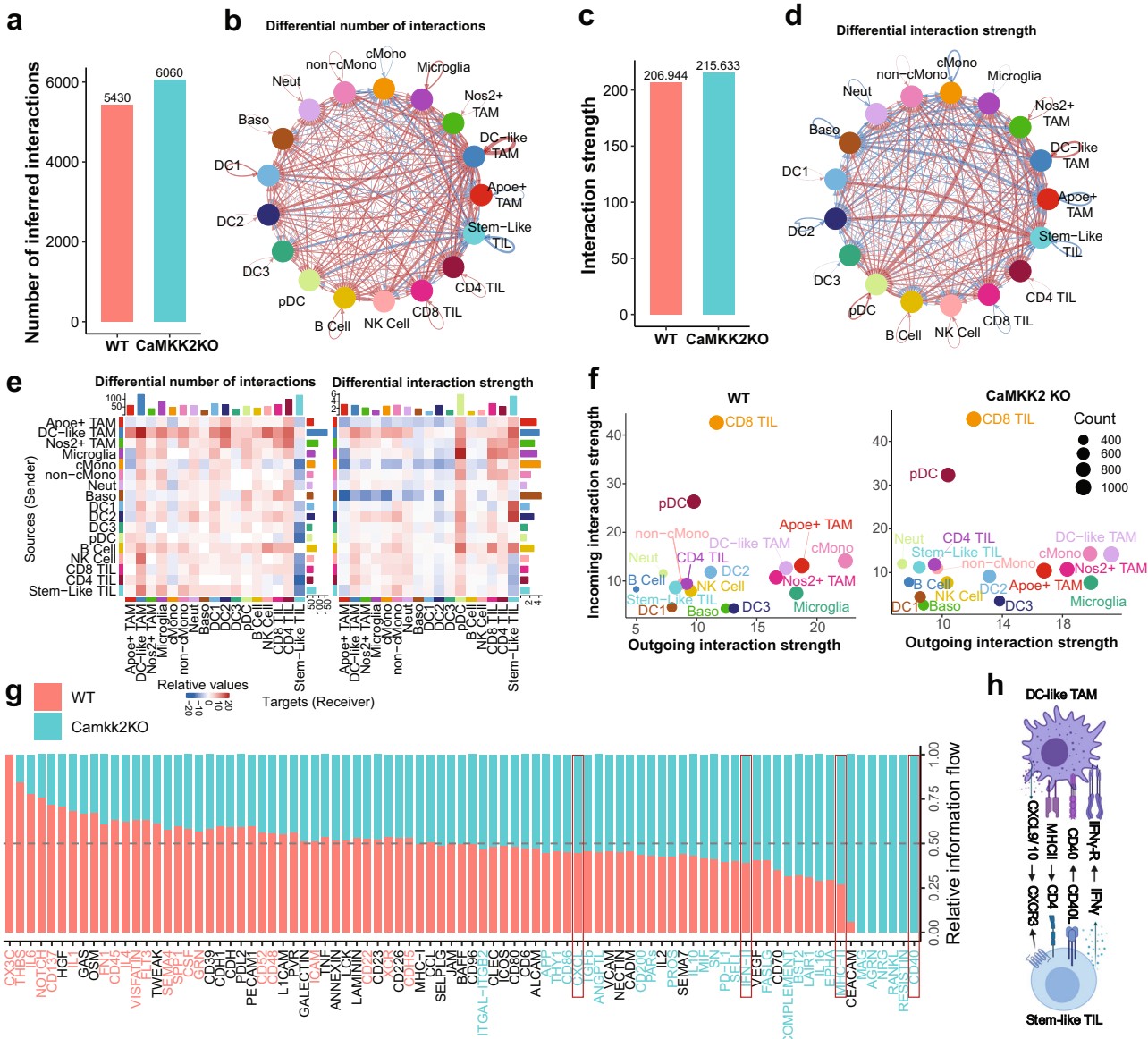

**Fig. 4 | Cell–cell interaction analysis infers communication between CD4+ stem-like T cells and DC-like TAMs in the setting of CaMKK2 deficiency. a, c** Cell–cell interaction analysis was performed on the scRNA-seq dataset using CellChat. Interaction strength represents the probability that a cell is sending or receiving a signal based on its expression of either ligands or cognate receptors, respectively. Total number of interactions and interaction strength of the inferred cell–cell communication networks for each genotype. **b, d** Differential number of interactions or interaction strength in the cell–cell communication network, visualized as a circle plot, where red or blue colored edges represent increased or decreased signaling in CaMKK2 KO compared to WT. Line thickness and darkness indicate the relative enrichment value. **e** Differential number of interactions or interaction strength, visualized as a heatmap, where red or blue represents increased or decreased signaling in CaMKK2 KO compared to WT. The top-colored bar plot represents the sum of a column of values displayed (incoming signaling). The right-colored bar plot represents the sum of a row of values (outgoing signaling). **f** Cell types with high outgoing interaction strength are expected to be initiators of cell–cell interactions, and cell types with high incoming interaction strength are expected to be targets of cell–cell interactions. Count refers to the number of inferred receptor-ligand pairs associated with each cell group. **g** Relative information flow from cell–cell interaction analysis. Receptor-ligand pathways with blue text are significantly enriched in CaMKK2 KO cells, and pathways with red text are significantly enriched in WT cells. **h** Summary diagram of cell interaction enriched in CaMKK2 KO mice. Graphic was created with BioRender.com.

deficiency licenses CD8 T-cell cytotoxicity and restricts terminal exhaustion.

## Cell–cell interaction analysis infers communication between CD4+ stem-like T cells and DC-like TAMs in the setting of CaMKK2 deficiency

To determine which other immune cells may be interacting to promote this ICB-responsive CD8+ T-cell phenotype, we performed cell–cell interaction analysis on the scRNA-seq data. Cell–cell interaction analysis infers intercellular communication using a database of interactions among ligands, receptors, and their cofactors. This allows for the identification of cell types that are likely initiators (high expression of ligand) and targets (high expression of the cognate receptor) of cell–cell interactions in the GBM TME. Cell–cell interaction analysis also provides insights into those interactions that are enriched in CaMKK2 KO mice, to help identify biomarkers and mechanistic determinants of ICB responsiveness and anti-tumor function. Overall, the number of predicted receptor-ligand interactions identified in CaMKK2 KO mice was higher, with a notable increase in identified interactions with DC-like TAMs (Fig. 4a, b, e).

The predicted differential interaction strength, which indicates the probability of cell types communicating more frequently in each genotype, inferred that stem-like TILs are more common targets of interactions in the CaMKK2 KO TME than in the WT TME (Fig. 4c–e). Stem-like TILs are predicted to be primarily acted upon by DC1s, DC2s, and DC-like TAMs (Fig. 4d, e). This indicates a role for these antigen-presenting cells in supporting the stem-like TILs. This analysis identified MNPs as likely initiators of interactions among immune cells in the TME, based on high predicted outgoing interactions strength (Fig. 4f). In particular, classical monocytes (cMonos), Apoe+ TAMs, and microglia were identified in WT mice, while cMonos, DC-like TAMs, and microglia were identified in CaMKK2 KO mice. CD8+ TILs were identified as targets of interactions, based on their predicted incoming interaction strength, regardless of genotype (Fig. 4f).

Information flow analysis enables the inference of cell–cell communication pathways that are conserved within a genotype. This technique predicted that CD40 (*CD40-CD40lg*), MHC-II (*H2-Aa-CD4*), IFN-II (*Ifng-Ifngr*), and CXCL (*Cxcl9-Cxcr3*, *Cxcl10-Cxcr3*) receptor-ligand interactions were significantly enriched in the CaMKK2 KO TME (Fig. 4g and Supplementary Fig. 3a). These pathways were predicted to be more active in CaMKK2 KO stem-like TILs and TAMs (Fig. 4h and Supplementary Fig. 3a), further suggesting greater interaction between CaMKK2-deficient TAMs and TILs. These data also demonstrate that MNPs (particularly DC-like and Apoe+ TAMs) likely play important roles in shaping the anti- and pro-tumor GBM immune TME via interactions with TILs.

## CaMKK2 deficiency promotes tumor infiltration by effector CD4+ T cells

Cell–cell interaction analysis indicated that stem-like TILs, particularly CD4+ stem-like TILs, were important targets of immune interactions in the GBM TME. This motivated us to further examine the reclustered scRNA-seq TIL data (Fig. 3a, b). The originally identified stem-like TIL population (Fig. 2c) contained appreciable heterogeneity. The newly identified effector CD4+ TILs and γδ TILs were almost entirely derived from the original stem-like TIL population (Supplementary Fig. 4a). This suggests that the identified cell–cell interactions between DC-like TAMs and stem-like TILs, such as those via CD40 (*CD40-CD40lg*) and MHC-II (*H2-Aa-CD4*) (Fig. 4h) may also occur amongst stem-like TILs, effector CD4+ TILs, and γδ TILs, as identified through reclustering.

When stratifying the UMAP projection of TILs by genotype (WT vs. KO), it became apparent that the effector CD4+ population (*Cd4, Cd40lg, Ifng, Tnf*) was almost exclusively present in CaMKK2 KO mice (Fig. 5a, b). Utilizing intracellular cytokine staining, we found CD40L+ IFNγ+ CD4+ TILs to be substantially enriched in the TME of CaMKK2 KO mice (Fig. 5c, d). CD40L+ TNFa+ and IFNγ+ TNFa+ CD4+ TILs were also enriched in CaMKK2 KO mice (Supplementary Fig. 4b, c). Th1 signatures, typically denoted by IFNγ and CD40L (as we observed), are known to be associated with a more anti-tumor TME phenotype and ICB responsiveness[6]. Considering this, we sought to determine if CD4+ T cells were also necessary for the improved survival observed in CaMKK2 KO mice, using depleting antibodies. Indeed, as with CD8+ T cells, CD4+ T cells were critical for the anti-tumor effect mediated by CaMKK2 deficiency (Fig. 5e).

Flow cytometric analysis additionally indicated that the relative frequency of CD4+ and CD8+ TIL populations were dramatically skewed toward CD4+ TILs in the CaMKK2 KO TME (Supplementary Fig. 4d). This shift reflected an increased absolute accumulation of CD4+ TILs, and not reduced numbers of CD8+ TILs (Supplementary Fig. 4d). Notably, the increased accumulation of CD4+ TILs in the CaMKK2 KO TME was not driven by an expansion in Treg abundance (Supplementary Fig. 4e). To further assess the intratumoral accumulation of CD4+ TILs in CaMKK2 KO mice, we utilized confocal microscopy. These experiments confirmed that there were significantly more CD4+ TILs per mm³ of tumor volume, but also that intratumoral penetrance was

significantly enhanced in CaMKK2 KO mice (Fig. 5f–h). This, in turn, suggests that CaMKK2 may restrict the abundance of CD40L+ IFNγ+ TNFa+ effector CD4+ TILs, the density of CD4+ cells per mm³ of tumor volume, and the tumor penetrance of CD4+ T cells.

As MHC-II-CD4 and CD40L-CD40 interactions were found to be enriched in the CaMKK2 KO immune TME via cell–cell communication analysis, and as we observed improved tumor penetrance of CD4+ T cells, we utilized confocal microscopy to determine if there were increased myeloid-CD4+ interactions in CaMKK2 KO mice. *Iba1* is highly and specifically expressed in myeloid cells in the TME (Supplementary Fig. 4f), so it was assessed here as a marker of intratumoral myeloid infiltration. Indeed, CD4-myeloid interactions were found to be more frequent in the CaMKK2 KO TME (Supplementary Fig. 4g–j). This is evidence that the immunostimulatory myeloid cells, and CD4+ TILs–which have an increased abundance of the Effector CD4+ phenotype–are co-operating with greater frequency in the CaMKK2 KO TME.

## Immunostimulatory phenotype emerges among TME mononuclear phagocytes in the absence of CaMKK2

Since DC-like and Apoe+ TAMs represented the largest phenotypic shifts in the scRNA-seq profiling of the immune TME and were found to be important for shaping the immune TME by cell–cell interaction analysis, we further analyzed how these two populations differed in anti-tumor function and promotion of ICB response. Consistent with their DC-like classification, DC-like TAMs were highly enriched for antigen processing and presentation via MHC-I and MHC-II by differential expression and gene ontology (GO) biological process analysis (Fig. 6a, b). TMEs that are enriched for MHC-I and MHC-II signatures are considered to possess greater anti-tumor activity[6] and are associated with superior responses to ICB[6,38]. *CD40* expression was upregulated on DC-like TAMs and had been already identified above as prominently involved in cell–cell communication analysis (Figs. 6a and 4h). Beyond antigen presentation, DC-like TAMs were also enriched for other anti-tumor biological process signatures, such as Type 1 and 2 Interferon response, induction of T-cell cytotoxicity, induction of T-cell migration, and induction of T-cell proliferation (Fig. 6b).

Of the genes that were upregulated in Apoe+ TAMs, *Apoe* was the most differentially expressed (Fig. 6a). Recently, Apoe has been identified as a hallmark gene for the DAM phenotype, which is present in various neurodegenerative diseases[39] and is associated with an ICB resistant TAM phenotype[9,10]. Interestingly, several of the genes associated with the DAM phenotype were found to be enriched in Apoe+ TAMs (*Apoe, Cd63, Trem2*, Spp1, *Lpl, Cd9*) (Fig. 6a). Projection of the DAM signature into UMAP space, with only the MNP system embedded, showed a strong colocalization of the DAM signature with the Apoe+ TAM and microglia cell clusters (Fig. 6c, d). Extending this analysis to the whole MNP system showed that DAM genes were consistently enriched in MNPs in the WT TME, while antigen processing, interferon (IFN) response and immunostimulatory genes were enriched in CaMKK2 KO MNPs (Fig. 6e and Supplementary Fig. 5a). Utilizing flow cytometry, we confirmed that MHC-II and CD40 were more highly expressed on CaMKK2 KO TAMs and cMonos (Fig. 6f, g). We additionally examined the ratio of immunostimulatory (MHC-II+, CD40+) to DAM-like (Mrc1+, Trem2+) MNPs and found that a favorable ratio was detected in CaMKK2 KO TAMs and cMonos (Fig. 6h–k).

Many macrophage-directed therapies in GBM have targeted macrophage survival and recruitment, but few therapies have demonstrated an impact on phenotypic programming without limiting macrophage accumulation in the TME[40]. Reprogramming TAMs in GBM is an attractive therapeutic strategy due to the abundance of TAMs in the TME and their potential for anti-tumor function[41]. To further validate the role of CaMKK2 in TAM programming, we utilized immunofluorescence and confocal microscopy. Confocal microscopy confirmed that the abundance of Apoe+ myeloid cells is significantly

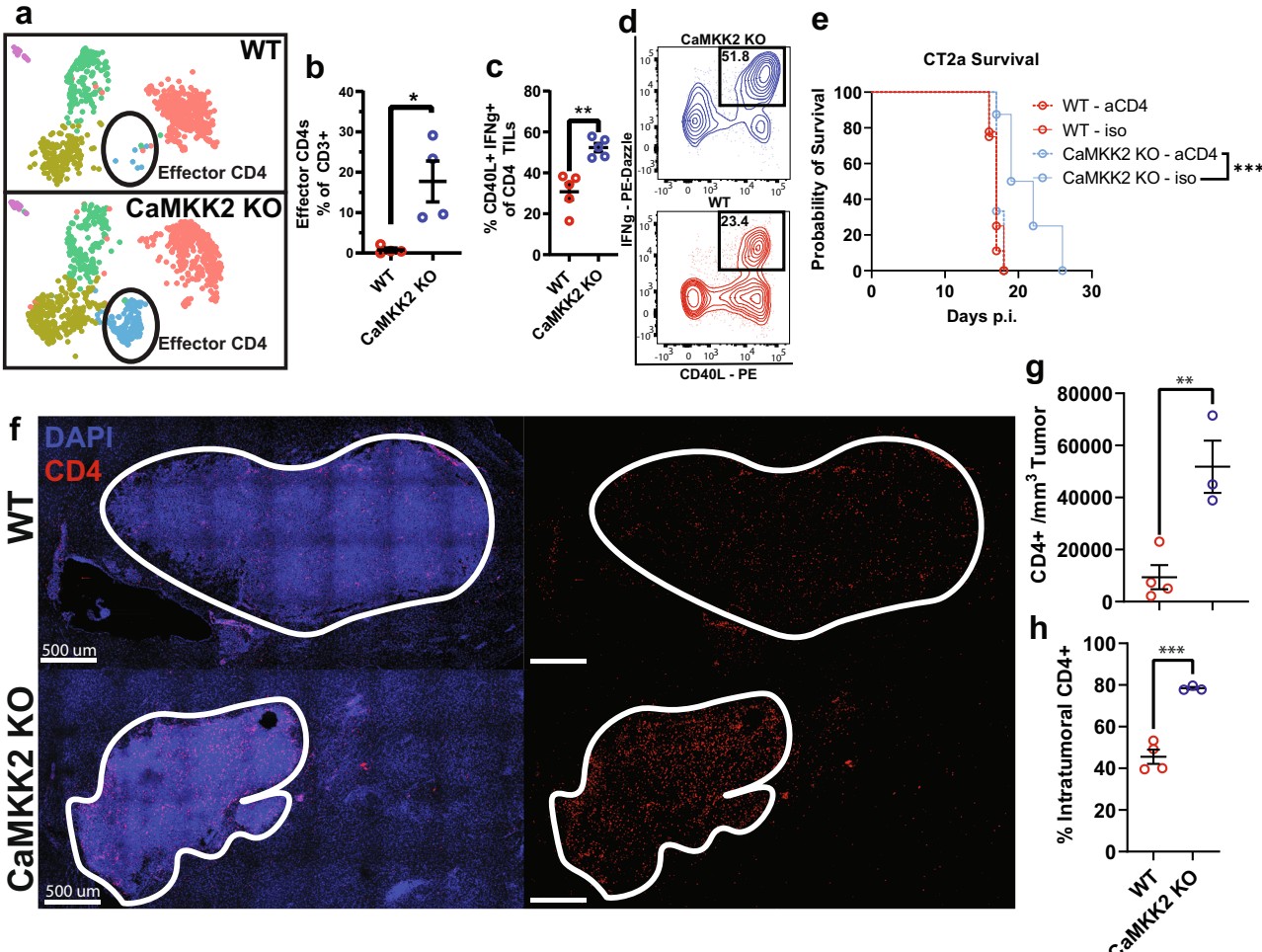

**Fig. 5 | CaMKK2 deficiency promotes tumor infiltration of effector CD4⁺ T cells.**
**a** UMAP plots of reclustered TILs stratified by genotype **b** Abundance of Effector CD4⁺ TILs as a percentage of reclustered TILs. $n = 4$ per genotype, two-tailed $t$-test. *$p = 0.0156$. **c, d** Tumor-bearing hemispheres were harvested on D14 post CT2a implantation and stained with a multi-color flow cytometry panel to assess abundance of CD4 IFNγ⁺ CD40L⁺ TILs. $n = 5$ per genotype, two-tailed $t$-test. **$p = 0.0015$. **e** 50k CT2a was implanted intracranially and mice were monitored for survival. Either 200 ug of isotype or aCD4 was administered on D-3,D-2,D-1, p.i. and every 3 days after through D14 for a total of 8 treatments. $n = 8$ per group, Log-rank test.

***$p = 0.0005$. **f** Representative images of D14 tumor-bearing hemispheres stained with CD4 and DAPI to identify CD4+ cells and nuclei, respectively. Experiment was independently repeated twice. **g** Number of CD4⁺ cells found per mm³ tumor by confocal microscopy images of D14 tumor-bearing hemispheres. $n = 4$ WT, $n = 3$ CaMKK2 KO, two-tailed $t$-test. **$p = 0.0082$. **h** Percentage of intratumoral CD4⁺ cells of total CD4⁺ cells identified, in D14 tumor-bearing hemispheres, by confocal microscopy. $n = 4$ WT, $n = 3$ CaMKK2 KO, two-tailed $t$-test. ***$p = 0.0005$.
**b, c, g, h** Data are presented as mean ± SEM. Source data are provided as a Source Data file.

reduced in tumors of CaMKK2 KO mice (Supplementary Fig. 5b, c). In addition, there was no significant reduction in myeloid infiltration per mm³ of the tumor in the setting of CaMKK2 deficiency, and there was instead a higher percentage of myeloid cells found intratumorally, rather than peritumorally (Supplementary Fig. 5d, e). Collectively, these results suggest that CaMKK2 promotes a DAM-like phenotype associated with ICB resistance in TAMs and other MNPs, which are the most abundant immune cells in the GBM TME (Fig. 2d and Supplementary Fig. 2b). Likewise, CaMKK2 may prevent TAMs from being programmed to an immunostimulatory phenotype, which may better promote ICB response.

**CaMKK2 deficiency in non-hematopoietic cells is necessary for licensing checkpoint blockade response and immunostimulatory macrophages**
Considering the high expression of CaMKK2 in TAMs (Fig. 1b and Supplementary Fig. 2d), TAM abundance in the TME (Fig. 2d and Supplementary Fig. 2b), and the immunostimulatory impact of CaMKK2 deficiency on TAMs, we conditionally deleted CaMKK2 in MNPs (LysM^cre x CaMKK2^fl/fl) to determine whether this would be

sufficient to produce a survival benefit in IC tumor-bearing mice. Although this conditional deletion model has induced tumor regression in preclinical breast cancer models[22], there was no observable survival benefit in orthotopic GBM (Fig. 7a). Furthermore, conditional deletion of CaMKK2 in MNPs was insufficient to induce the MHC-II^high phenotype in MNPs that was observed in germline CaMKK2 KO mice (Fig. 7b). Considering that CaMKK2 is highly expressed in both hematopoietic and non-hematopoietic cells, we utilized bone marrow chimeras to determine which compartment was primarily mediating the impact of CaMKK2 deficiency.

In a reciprocal bone marrow chimera model, where irradiated WT and CaMKK2 KO mice received either WT or CaMKK2 KO bone marrow, respectively, (Supplementary Fig. 6a, b), we demonstrated that CaMKK2 deficiency in non-hematopoieticcells was necessary for any survival benefit (Fig. 7c). However, survival was further enhanced by the combination of CaMKK2 deficiency in both the hematopoietic and non-hematopoietic compartments. The ultimate requirement for CaMKK2 deficiency in non-hematopoietic cells, however, suggests a strong role for CaMKK2 in brain-native cells, which interact with immune cells within the TME. Bone marrow chimera models

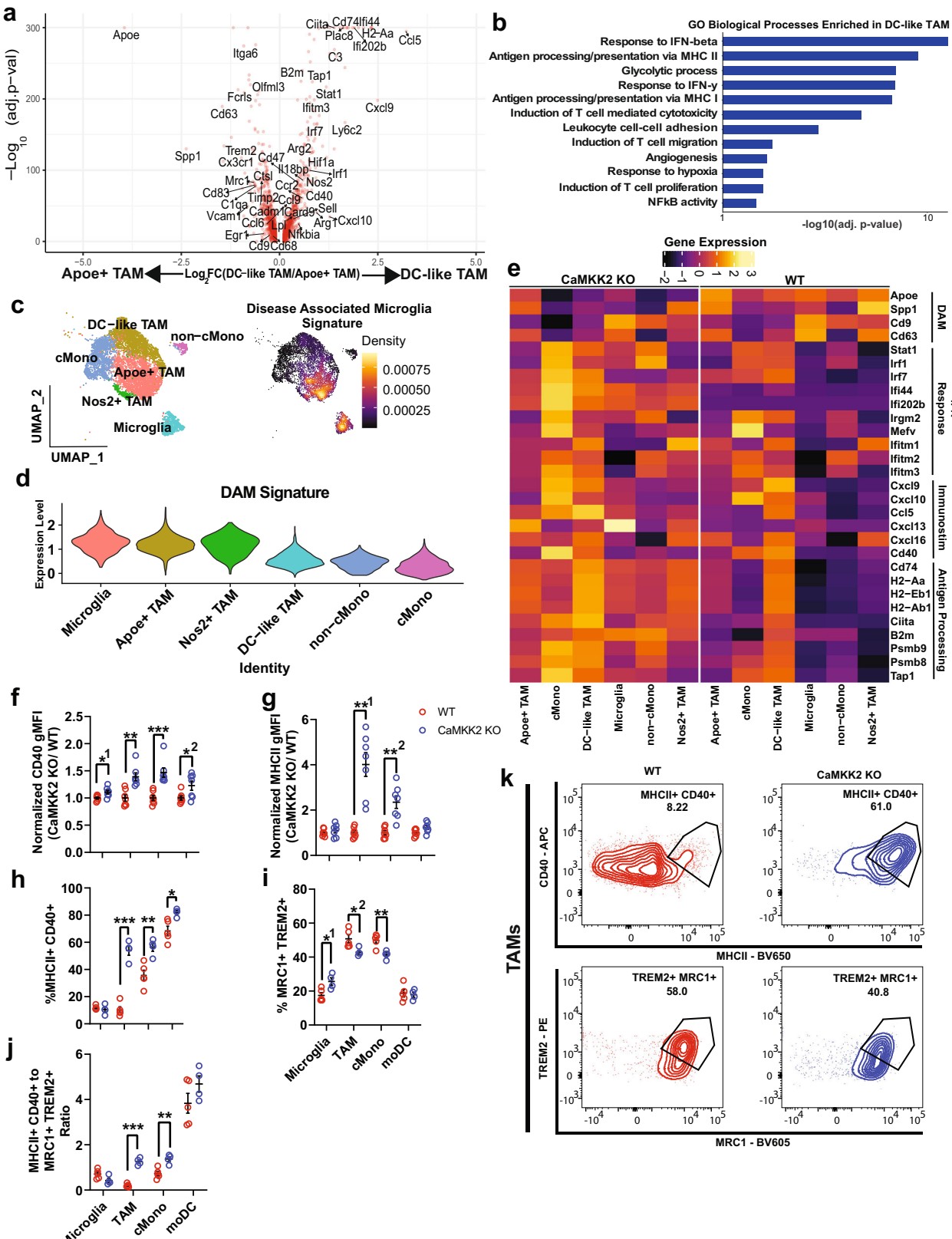

additionally identified that CaMKK2 deficiency in non-hematopoietic cells is necessary for ICB response (Fig. 7d). These results demonstrate a role for CaMKK2 in non-hematopoietic cells as primary drivers of ICB resistance in GBM.

To determine whether hematopoietic or non-hematopoietic cells were also responsible for increasing the prevalence of MHC-II[high] TAMs

in tumors of CaMKK2 KO mice, we again utilized a bone marrow chimera model. Surprisingly, CaMKK2 deficiency in non-hematopoietic cells was again found to be necessary for eliciting the MHC-II[high] TAM phenotype (Fig. 7e, f). Additional chimera experiments did however indicate that non-hematopoietic CaMKK2 deficiency was disposable for the CD4[+] TIL skewing, and reduced CD8[+] TIL exhaustion shown in

**Fig. 6 | Immunostimulatory phenotype emerges among TME mononuclear phagocytes in the absence of CaMKK2. a** Volcano plot of genes differentially expressed between DC-like and Apoe+ TAMs. Differential expression testing was performed using MAST and only genes with unadjusted $p$ values < 0.05 are shown. $n$ = 2276 Apoe⁺ TAMs $n$ = 1959 DC-like TAMs. **b** Differentially expressed genes from panel **a** were used in GO biological processes enrichment for processes enriched in DC-like TAMs relative to Apoe+ TAMs. **c** UMAP plot of MNPs and Density plot of the Disease Associated Microglia Signature projected in UMAP space. **d** Violin plot of DAM signature expression levels across mononuclear phagocytes. **e** Heatmap of genes related to the DAM phenotype, interferon response, chemotaxis, and antigen processing and presentation. **f, g** Tumor-bearing hemispheres were harvested on D14 post CT2a implantation from WT, CaMKK2 KO mice and stained with a multi-color flow panel to assess MHC-II and CD40 expression. $n$ = 7 per genotype, two-way RM ANOVA $p$ < 0.05 with post hoc unadjusted two-tailed Fisher LSD $t$-test. Each sample was normalized to the average WT gMFI. **f** *¹$p$ = 0.0198, **$p$ = 0.0012, ***$p$ = 0.0009, *¹$p$ = 0.038. **g** **¹$p$ = 0.0011, **²$p$ = 0.0022. **h**–**k** Tumor-bearing hemispheres were harvested on D14 post CT2a implantation from WT, CaMKK2 KO mice and stained with a multi-color flow panel to assess ratios of immunostimulatory and DAM-like MNPs. $n$ = 5 WT and $n$ = 4 CaMKK2 KO, two-way RM ANOVA $p$ < 0.05 with post hoc unadjusted two-tailed Fisher LSD $t$-test. **h** ***$p$ = 0.0001, **$p$ = 0.0037, *$p$ = 0.0149. **i** *¹$p$ = 0.0246, *²$p$ = 0.0263, **$p$ = 0.005. **j** ***$p$ = 0.0002, **$p$ = 0.0029. **f**–**j** Data are presented as mean ± SEM. Source data are provided as a Source Data file.

Supplementary Fig. 4d and Fig. 3g (Supplementary Fig. 6b, c). This indicates that hematopoietic CaMKK2 deficiency is sufficient to promote these anti-tumor TIL phenotypes. Focusing on the relevant brain-native non-hematopoietic cells, recent research suggests that neurons can have profound pro-tumor effects within the glioma TME via secretion of CaMKK2-dependent pro-tumor mitogenic factors[25–28]. Likewise, we found CaMKK2 to be frequently expressed in neurons (Fig. 1a and Supplementary Fig. 1a–c). To determine, then, if it was neuronal CaMKK2 deficiency alone that was sufficient to induce the MHC-IIʰⁱᵍʰ TAM phenotype, we conditionally deleted CaMKK2 in neurons using a Syn1ᶜʳᵉxCaMKK2ᶠˡ/ᶠˡ model. Indeed, conditional deletion of CaMKK2 in neurons was sufficient to induce the MHC-IIʰⁱᵍʰ TAM phenotype (Fig. 7g, h). Furthermore, deletion of CaMKK2 in neurons was also sufficient to improve survival and ICB response (Fig. 7i, j). This implicates neuronal CaMKK2 as a key contributor to the tumor-promoting TAM phenotype, tumor progression, and ICB resistance within the GBM TME.

## Discussion

Here, we applied a systems biology approach to survey the differences in the immune landscape of the WT and CaMKK2-deficient TME. Using this approach, we identified that CaMKK2 deficiency in neurons is sufficient to induce ICB response and anti-tumor immune phenotypes. Germline CaMKK2 deficiency increases the expression of cytotoxic molecules in CD8⁺ TILs and limits their exhaustion. Given that CD4⁺ T cells were critical for the survival benefit in CaMKK2-deficient mice and that these mice demonstrated increased intratumoral accumulation of an effector CD4⁺ phenotype (associated with ICB response), we suspect that CD4⁺ TILs additionally strongly contribute to the overall anti-tumor effect and ICB response in the CaMKK2 KO TME. Cell–cell interaction analysis indicated that CD4⁺ TILs were more frequently interacting with a MHC-IIʰⁱᵍʰ DC-like TAM phenotype, which was confirmed by microscopy. These DC-like TAMs were found to be primarily in the CaMKK2-deficient TME and were enriched for immunostimulatory transcriptional programs. Conversely, Apoe+ TAMs were primarily detected in the WT TME and were reminiscent of the DAM phenotype, which is associated with ICB resistance. Using bone marrow chimera models, we identified that CaMKK2 in the non-hematopoietic compartment was primarily responsible for driving ICB resistance. This led to our identification of neurons as important in maintaining an MHC-IIˡᵒʷ TAM phenotype, promoting tumor progression, and stimulating ICB resistance. CaMKK2's pro-tumor effects are likely mediated by multiple cell types, including immune and brain-native cells. This is unsurprising given the near-ubiquitous expression of CaMKK2 and the complexity of the TME. Indeed, our bone marrow chimera data indicate that hematopoietic CaMKK2 deficiency becomes relevant in the context of non-hematopoietic CaMKK2 deficiency in terms of impacting tumor progression. It is worth noting here that we observed that the chimera generation process, as well as iso-type antibody treatment, had negative effects on survival, which generally led to smaller—although still significant—differences in survival compared to untreated survival studies. In the CT2a preclinical model,

the combination of CaMKK2 deficiency along with ICB treatment approximately doubles median survival which we believe is suggestive that CaMKK2 may be a clinically significant target in GBM.

Collectively, these observations led us to construct our working model (Fig. 8). Neuronal CaMKK2 has profound pro-tumor effects, demonstrated by its ability to maintain TAMs in an ICB resistance-associated phenotype, as well as promote tumor progression and ICB resistance. CaMKK2 within the hematopoietic additionally has pro-tumor effects as evidenced by the bone marrow chimera experiments. Figure 7c indicates that hematopoietic CaMKK2 deficiency extends survival in the context of non-hematopoietic CaMKK2 deficiency. In addition, Supplementary Fig. 6b, c suggests that CaMKK2 in the hematopoietic compartment is sufficient to phenocopy TIL phenotypes that were observed in the germline CaMKK2 KO mice. We hypothesize that neuronal CaMKK2 is a primary driver of ICB-resistance via maintaining TAMs, an abundant component of the GBM TME, in a pro-tumor phenotype. The pro-tumor TIL phenotypes seem to be driven by hematopoietic CaMKK2 expression, likely as a combination of expression in innate and lymphoid cells. How CaMKK2 in neurons is interacting with the immune system is a topic of great interest. CaMKK2 in neurons likely has direct and indirect immuno-suppressive effects. Neuronal CaMKK2 may indirectly influence immunosuppression via supporting tumor growth through the secretion of CaMKK2-dependent mitogenic factors such as BDNF. These larger, faster growing, tumors would be expected to exert stronger immunosuppressive effects. Alternatively, neuronal CaMKK2 may be directly immunosuppressive through neuro-immune interactions. Collectively these results indicate that CaMKK2 promotes tumor progression, ICB-resistance, and pro-tumor immune phenotypes.

This work has translational implications for how CaMKK2 inhibition may be particularly efficacious in GBM due to the abundance of cells with pro-tumor functions which highly express CaMKK2, like neurons and TAMs. Because CaMKK2 appears to have pro-tumor functions in human GBM, and deletion of CaMKK2 extends survival in preclinical models, we expect that a brain penetrant CaMKK2 inhibitor may be efficacious as a monotherapy. Unfortunately, commercially available CaMKK2 inhibitors are neither very selective[42] nor brain penetrant[43]. Therapeutic targets which selectively re-polarize the stromal elements of the TME to anti-tumor phenotypes and enable ICB therapy have been limited, particularly in immunologically cold tumors such as GBM. CaMKK2 has roles in mediating ICB resistance. Thus, we expect a combination therapy of a CaMKK2 inhibitor and ICB would be more effective than either treatment alone in GBM. The immunostimulatory program that is active in CaMKK2-deficient TAMs likely contributes to ICB response and is quite unique among myeloid-directed therapeutic targets. Initially, myeloid targeted therapies aimed to simply deplete myeloid cells via targeting survival or chemotaxis (aCSF1R, aCCL2, aCCR2)[41]. Subsequent generations of myeloid-directed therapies aimed to re-educate TAMs (aCD47, BLZ945), but resulted in TAM phenotypes that were not directly immunostimulatory, and thus were not as well-positioned to synergize with T-cell dependent immunotherapies[41]. Because CaMKK2 deficiency

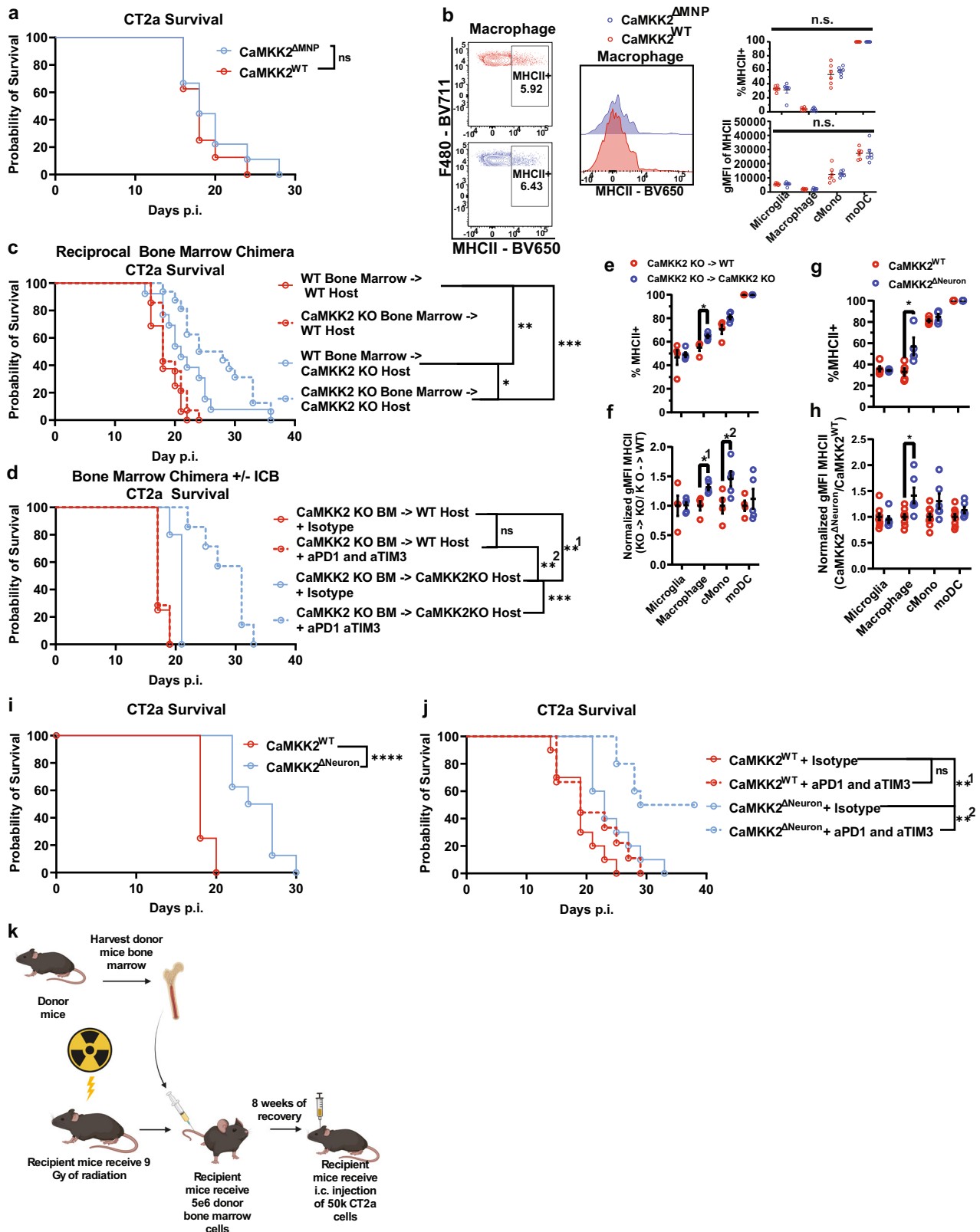

licenses such an immunostimulatory TME landscape, we anticipate it will also synergize with other T-cell-dependent therapies, such as chimeric antigen receptor (CAR)-T cells and tumor vaccines.

We have additionally demonstrated mechanisms for non-hematopoietic CaMKK2 in driving ICB resistance, as well as for neuronal CaMKK2 in maintaining a suppressive TME. These results

suggest that neuronal CaMKK2 deficiency can effectively program the tumor-infiltrating myeloid population from a DAM-like phenotype (associated with ICB resistance) to a more immunostimulatory phenotype. These findings represent an interface between immunotherapy and cancer neuroscience in GBM. Furthermore, they suggest that CaMKK2 has roles in maintaining the DAM phenotype,

**Fig. 7 | CaMKK2 deficiency in neurons is sufficient for licensing checkpoint blockade response and immunostimulatory macrophages. a, c, d, i, j** 50k CT2a was implanted and mice were monitored for survival, Log-rank test. **b, e–h** Tumor-bearing hemispheres were harvested on D14 post 50k CT2a implantation, and assessed by flow cytometry. 2-way RM ANOVA $p < 0.05$ with post hoc unadjusted two-tail $t$-test. Data are presented as mean ± SEM. **k** Schematic depicting bone marrow chimera generation. Graphic was created with BioRender.com. **c, d** Bone marrow chimeras were generated as shown in panel **k. d, j** ICB treatment regimen is described in panel **f. a** $n = 8$ per genotype (LysMcre$^{WT}$xCaMKK2$^{fl/fl}$, or LysMcre$^{+/-}$xCaMKK2$^{fl/fl}$). **b** $n = 6$ per genotype (LysMcre$^{WT}$xCaMKK2$^{fl/fl}$, or LysMcre$^{+/-}$xCaMKK2$^{fl/fl}$). **c** $n = 16$ WT->WT, $n = 14$ CaMKK2 KO->WT, $n = 13$ WT->CaMKK2 KO, $n = 16$ CaMKK2 KO->CaMKK2 KO, results are combined from two experiments. \*\*\*$p < 0.0001$, \*\*$p = 0.0084$, \*$p = 0.0467$. **d** $n = 8$ KO->WT + isotype, $n = 9$ KO->WT + ICB, $n = 7$ KO->KO + isotype, $n = 9$ KO->KO + ICB. \*\*$^1p = 0.0082$, \*\*\*$p = 0.0009$, \*\*$^2p = 0.0030$. **e, f** $n = 4$ KO->WT, $n = 5$ KO->KO. **e** \*$p = 0.0313$. **f** Each sample was normalized to the average WT gMFI. \*$^1p = 0.0213$, \*$^2p = 0.0438$. **g** $n = 4$ for Syn1cre$^{+/-}$xCaMKK2$^{fl/fl}$ and $n = 5$ for Syn1cre$^{WT}$xCaMKK2$^{fl/fl}$, \*$p = 0.0256$. **h** $n = 6$ for Syn1cre$^{+/-}$xCaMKK2$^{fl/fl}$ and $n = 9$ for Syn1cre$^{WT}$xCaMKK2$^{fl/fl}$. Combined from two experiments. Each experimental replicate was normalized to the average WT gMFI. \*$p = 0.0106$. **i** $n = 8$ per genotype (Syn1cre$^{+/-}$xCaMKK2$^{fl/fl}$ and Syn1cre$^{WT}$xCaMKK2$^{fl/fl}$), \*\*\*\*$p < 0.0001$. **j** $n = 10$ per condition except $n = 9$ for Syn1cre$^{WT}$xCaMKK2$^{fl/fl}$ + ICB group, \*\*$^1p = 0.0054$, \*\*$^2p = 0.0023$. Source data are provided as a Source Data file.

which likely has implications in other neurodegenerative diseases, like Alzheimer's.

Neuroimmunology has unveiled a myriad of neuro-immune interactions, many of which take place along vasculature that shuttles circulating immune cells into the TME. The perivascular niche, which is maintained by microglia and serves as a reservoir for neural and glioma stem cells, is a point of entry for many immune cells that go on to adopt pro-tumor phenotypes[3,4]. The permeability of vasculature in the central nervous system (CNS) is partially regulated by IFNγ secretion by CD4$^+$ T cells[44]. The increased abundance of IFNγ secreting CD4$^+$ TILs in CaMKK2-deficient mice, in addition to chemotactic signals from DC-like TAMs (*Cxcl9, Cxcl10*), may explain the enhanced tumor penetrance seen by CD4$^+$ and myeloid cells in the setting of CaMKK2 deficiency. How these myeloid cells are polarized to immunostimulatory or DAM-like phenotypes by neurons will be a topic of future study.

Neurons are known to establish intimate relationships with macrophages throughout the body[45] and may be directly polarizing TAMs to a pro-tumor phenotype in GBM. Whether this is the case could be informed by comparative sequencing of WT and CaMKK2-deficient neurons in the GBM TME. Alternatively, pro-tumor macrophage programming may take place via an indirect mechanism, such as by neurons influencing the immunosuppressive capacity of tumors through the secretion of neurotrophic factors like BDNF, as previously mentioned. In addition to neurons and TAMs, cancer-associated fibroblasts have been identified as having a pro-tumor CaMKK2-dependent role in a pancreatic cancer model[46] and are a therapeutic target in other cancers[41]. Although fibroblasts are not traditionally thought of as part of the GBM TME, recent work has identified that they play crucial roles in fibrosis after injury in the CNS[47]. These may represent an additional non-hematopoietic CaMKK2-expressing cell with pro-tumor roles in GBM.

In summary, we found that CaMKK2 deficiency dramatically remodels the immune TME to a more anti-tumor, ICB-responsive phenotype, and away from phenotypes associated with ICB resistance. These studies also lay the foundation for an area of research on the immunosuppressive effects of neurons in the GBM TME and demonstrate that therapeutic inhibition of CaMKK2 may prolong survival in patients with GBM, as well as improve the effectiveness of ICB in this setting.

## Methods
### Mice
Six- to eight-week-old C57BL/6J, LysM$^{cre}$, Syn1$^{cre}$, and CD45.1 mice were purchased from the Jackson Laboratory. CaMKK2$^{-/-}$, Tg(Camkk2-EGFP) DF129Gsat reporter mice (CaMKK2-EGFP), and CaMKK2$^{fl/fl}$ mice were generously provided by Luigi Racioppi (Duke University). CaMKK2$^{-/-}$, CaMKK2-EGFP and CaMKK2$^{fl/fl}$ mice have been previously validated[22]. All transgenic mouse lines were derived from or have been previously backcrossed to the C57BL/6 background. Animals were maintained under pathogen-free conditions, in temperature and humidity controlled housing, with free access to food and water, under a 12-h light/dark cycle at the Cancer Center Isolation Facility of Duke University Medical Center. All experimental procedures were approved by the Institutional Animal Care and Use Committee at Duke University Medical Center.

### Cell lines
C57BL/6 syngeneic CT2a, GL261, and KR158B-Luc were provided by Robert L. Martuza (Massachusetts General Hospital), the National Cancer Institute, and Duane Mitchell (University of Florida), respectively. CT2a and KR158B-Luc cells were cultured in complete DMEM (Gibco 11995-065, 10% FBS). GL261 cells were cultured in complete RPMI (Gibco 11875-093, 1% Non-essential amino acids, 1% Sodium Pyruvate, 10% FBS). For intracranial implantation, $5 \times 10^4$, $1 \times 10^5$, $5 \times 10^4$ cells were implanted for CT2a, GL261, and KR158B-Luc respectively. For subcutaneous implantation, $2.5 \times 10^5$ CT2a cells were injected into the flank. All cell lines were authenticated and tested negative for mycoplasma, and interspecies contamination by IDEXX Laboratories.

### Tumor inoculation
Tumor cells were collected in their logarithmic growth phase via trypsinization (Gibco 25300-054) and then resuspended in PBS. For intracranial implantation, tumor cells were mixed 1:1 with 3% methylcellulose and loaded into a 250 μl Hamilton syringe (Hamilton Company, 81120). Mice were anesthetized using isoflurane. Injection sites were shaved, then mice were placed in a stereotactic frame. After sterilization of the scalp, a midline incision was made to expose the bregma. The Hamilton syringe was positioned over the bregma, moved 2 mm laterally to the right, lowered 5 mm below the surface of the skull, and then raised 1 mm to create a pocket for the tumor suspension. An infusion pump was then used to infuse 5 μl of tumor cells at 120 μl min$^{-1}$ containing $5 \times 10^4$, $1 \times 10^5$, $5 \times 10^4$ cells for CT2a, GL261, and KR158B-Luc, respectively. After completion of the infusion, the syringe was left in place for an additional 45 s before removal. Bone wax was used to cover the injection site, and then the incision was stapled close. Mice were euthanized if there was any bulging of the skull or eyes or if they experience a failure to ambulate.

For subcutaneous tumor inoculation, tumor cells were harvested as described above and resuspended in PBS. Using a 1-ml syringe with a 25 G needle, $2.5 \times 10^5$ CT2a cells were implanted in the flank in 100 μl. A maximal tumor burden of 2000 mm$^3$ was not exceeded.

### In vivo antibody administration
For anti-CD4 and anti-CD8 depletion, 200 μg anti-CD4 (GK1.5, Bio X Cell), 200 μg anti-CD8 (2.43, Bio X Cell), or 200 μg isotype (LTF-2, Bio X Cell) were administered intraperitoneally in 500 μl PBS on the three days before tumor implantation. Depletion was maintained by treating every 3 days, starting day 2 post implantation, through day 14 for a total of eight treatments.

For immune checkpoint blockade experiments, either ICB combination therapy containing 200 μg anti-PD1 (RMP1-14, Bio X Cell) and 200 μg anti-TIM3 (RMT3-23, Bio X Cell) or 400 μg isotype (2A3, Bio X Cell). Injections were administered intraperitoneally in 500 μl PBS every 3 days, starting day 3, through day 18 post tumor implantation for a total of 5 treatments.

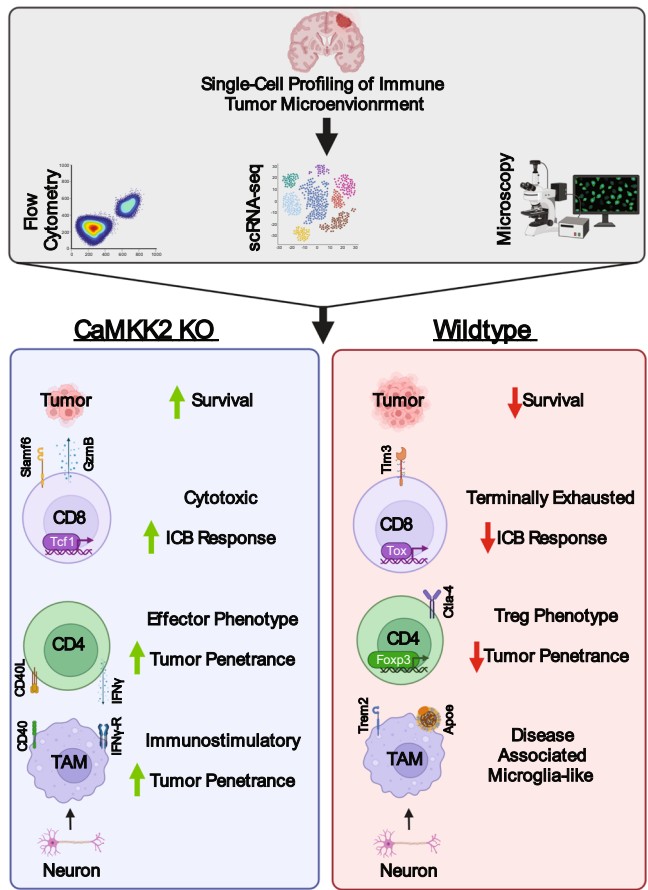

**Fig. 8 | Working model of CaMKK2's mechanism of action for promoting immunotherapeutic resistance.** Graphical summary of findings. Green and red arrows refer to the increased or decreased presence of a phenotype, respectively. Graphic was created with BioRender.com.

## Tissue processing and flow cytometry

The flow cytometry protocol is described in detail elsewhere[48]. In brief, after transcardial perfusion with PBS tumor-bearing hemispheres were harvested on day 14 post tumor implantation. Tissue was transferred to a Dounce Tissue Homogenizer with 5 ml of digestion cocktail containing 0.05 mg ml$^{-1}$ Liberase DL (Roche), 0.05 mg ml$^{-1}$ Liberase TL (Roche), 0.2 mg ml$^{-1}$ Dnase I (Roche) in HBSS with calcium and magnesium. A cell suspension was obtained after 5–10 strokes with the loose-fitting (A-size) pestle. The cell mixture was then incubated at 37 °C for 15 min in a water bath to obtain a single-cell suspension. The single-cell suspension was then passed through a 70-μm filter. After centrifugation, the cells were resuspended in 1× RBC Lysis Buffer (Thermo Fisher Scientific) for 3 min. Myelin was removed from the sample with Percoll centrifugation. Samples were centrifuged and mixed with 30% Percoll (Sigma Aldrich) and centrifuged at 500 g for 20 min at 18 °C with no brake. The myelin layer and Percoll were then aspirated and the pellet was resuspended in PBS before counting on an automated cell counter (Thermo Fisher Scientific).

For cytokine re-stimulation, samples are resuspended in 1 ml RPMI + 10% FBS at 1–2 × 10$^7$ cells ml$^{-1}$ in a 24 well plate, before viability, extracellular, or intracellular staining. 2 μl Cell Activation Cocktail with Brefeldin A (Biolegend) and 2 μl GolgiStop (BD Biosciences) were added and then samples were incubated for 4 h at 37 °C.

If samples were not stimulated, cells were then resuspended at 1–2 × 10$^7$ ml$^{-1}$ in 100 μl PBS and transferred to a 96-well plate. Before further staining, samples were resuspended in Zombie Aqua Viability Dye (1:400, Biolegend) and incubated for 30 min on ice.

For extracellular staining, samples were incubated with a blocking solution containing 2% Normal Rat Serum (Thermo Fisher Scientific), 2% Normal Armenian Hamster Serum (Innovative Research), 2% Normal Mouse Serum (Thermo Fisher Scientific), and TruStain FcX Plus (Biolegend) in MACS Buffer (PBS + 1% FBS + 1 mM EDTA) for 15 min on ice. After blocking, samples were incubated with antibodies (Supplementary Table 1) for 30 min on ice. Stained samples were then fixed in 2% formaldehyde in PBS on ice for 15 min.

For intracellular staining, samples were stained for viability and extracellular markers as described above. After staining, cells were fixed with 1× of FOXP3 Fixation/Perm buffer (eBioscience FOXP3/Transcription Factor Staining Buffer Set, Thermo Fisher Scientific) for 30 min on ice. Following fixation, samples were resuspended in 1× FOXP3 Perm/Wash buffer for overnight permeabilization at 4 °C. Intracellular antigens were then stained on ice for 30 min in 1× perm buffer.

Before acquisition, 10 μl of Accucheck Counting Beads (Thermo Fisher Scientific) were added to each sample. To calculate the number of cells per gram of tumor the following calculation was used: number of acquired cells × (number of input beads/number of acquired beads) × (1 / fraction of sample stained) × (1 / tumor weight). Samples were acquired on an LSRII (BD Biosciences) using FACS Diva software v.9 (BD Biosciences) and analyzed using FlowJo v.10 (Tree Star).

## Multiplexing samples by cell-hashing and flow sorting

Samples were processed as described above, and after viability and extracellular staining, each biological replicate for each genotype was stained with four unique oligo-tagged TotalSeqB (Biolegend) antibodies. Samples were incubated for 30 min on ice. Samples were then sorted on a FACSAria II (BD Biosciences) and enriched for CD45+ Live cells.

## scRNA-seq library preparation

Sorted live CD45$^+$ tumor-infiltrating cells were encapsulated into droplets and libraries were prepared using a Chromium Single Cell 3′ Kit using the v3.1 chemistry with feature barcoding (10X Genomics). For each 10X channel, 4 samples were equally combined: $n = 4$ WT, and $n = 4$ CaMKK2$^{-/-}$. A total of 7000 cells per genotype were targeted with 1750 cells per biological replicate contributing to each genotype. The generated scRNA-seq and hashtag libraries were pooled at a 25:1 ratio and sequenced on a Novaseq S Prime Flow Cell to an average depth of 61,286 and 59,013 reads per cell for WT and CaMKK2$^{-/-}$ samples, respectively. The WT library was sequenced to 56.2% saturation and a median of 3233 genes per cell were detected. The CaMKK2$^{-/-}$ library was sequenced to 59.6% saturation and a median of 2921 genes per cell were detected.

## scRNA-seq data analysis

The Cell Ranger pipeline (10X Genomics) was used to perform sample demultiplexing and to generate FASTQ files for the gene expression and hashtag libraries. FASTQ files were demultiplexed from the raw sequencing reads (bcl2fastq, v2.20), aligned to the mouse mm10 reference genome (cellranger, v4.0.0), and raw gene count matrices were generated using STAR (v2.7.5c).

Downstream analysis was performed using the R software Seurat package[49] (v4.0.3, http://satijalab.org/seurat/). Hashtag oligos, which corresponded to biological replicates, were demultiplexed using the HTODemux function and appended to the meta-data for each sample Low-quality cells, expressing less than 200 genes, and genes expressed by fewer than three cells were removed. The gene expression matrix for each genotype was then concatenated using the merge function in Seurat. The percentage of mitochondrial gene content was calculated using the Mouse.MitoCarta3.0[50] gene set and the PercentageFeatureSet function in Seurat. The Seurat object was converted to a SingleCellExperiment object, and outlier exclusion was performed in scater

(v.1.18.6). Using the isOutlier function in scater, cells were discarded if their percentage of mitochondrial gene content, number of expressed genes, or number of reads for a given cell was considered an outlier. A total of 11,784 Live CD45[+] cells passed QC and the SingleCellExperiment object was then converted back into a Seurat object. Normalization and regression of cell cycle scoring and percent mitochondrial gene content were performed using the SCTransform function in Seurat. Principal component analysis (PCA) was performed on all genes, and the number of principal components to be utilized in further analysis was determined heuristically using the elbow method. Thirty principal components were used for clustering and dimensionality reduction using FindNeighbors, RunUMAP, and a resolution of 0.5 was used for FindClusters in Seurat. This approach identified Seventeen distinct cell clusters. The RNA assay in the Seurat object was then multiplied by 10,000 and log-transformed before cluster-specific genes were identified. FindConservedMarkers was used to identify marker genes that were conserved between genotypes for each cell cluster. Annotation of cell clusters was performed by utilizing public datasets which identified marker genes for the identified cell types. Microglia signatures were obtained from Bowman et al. and Haage et al.[51,52]. Macrophage signatures were obtained from Gautier et al.[53]. All other signatures were obtained from a dataset that identified conserved cell types in mouse and human tumors[54]. Differential expression analysis, between cell types or genotypes, was performed using the MAST statistical test in the FindMarkers function in Seurat. Gene expression, gene signature scores, and clustering results were all visualized by embedding cells in dimensionally-reduced Uniform Manifold Approximation and Projection (UMAP) space. Gene signature scores were calculated using the AddModuleScore function in Seurat, and then a density plot was projected into UMAP space using the Nebulosa (v.1.0.2) package.

### Publicly available scRNA-seq data analysis
Publicly available scRNA-seq datasets (GSE8446, https://portal.brain-map.org/atlases-and-data/rnaseq/human-m1-10x), which had been processed and annotated previously[55,56], were imported into Seurat and processed using the workflow described above, with the exception that the annotations provided with the datasets were used. Expression of CaMKK2 was then visualized in healthy and tumor-bearing human tissues using Seurat's VlnPlot and Nebulosa's plot_density functions.

### Cell–cell communication analysis
Cell–cell communication analysis was performed using the CellChat[57] (v.1.0.0) software package in R. CellChat quantitatively infers and analyzes intercellular communication networks from scRNA-seq data using a curated database of known interactions among ligands, receptors, and their cofactors. Inference of cell type-specific signaling communication is performed using mass action models. The visualizations were created using the following vignette: https://htmlpreview.github.io/?https://github.com/sqjin/CellChat/blob/master/tutorial/Comparison_analysis_of_multiple_datasets.html.

### Functional analysis using gene ontology enrichment analysis
To predict putative biological functions based on differential gene expression, we performed a Gene Ontology (GO) analysis. Genes that were differentially expressed between Apoe[+] and DC-like TAMs (adjusted $p$ value <0.05, $\log_2$(FC) < 0.1) were inserted into the DAVID[58] functional annotation tool (https://david.ncifcrf.gov/tools.jsp). GOTERM_BP_DIRECT results were exported, shortlisted, and visualized in R. The full list of pathways are provided (Supplementary Data 6).

### Bone marrow chimera generation
Bone marrow was harvested from age (8–10 weeks) and sex-matched donor mice. Recipient mice received whole-body irradiation with a 9 Gy dose from a Cesium irradiator (Mark I-68A [137]Cs irradiator, JL Shepherd and Associates). Recipient mice then received an intravenous infusion of $5 \times 10^6$ donor bone marrow cells in 100 μl PBS. Recipient mice were then put on antibiotic-treated water for 2 weeks post bone marrow transfer. Donor chimerism was found to be nearly 100% in the bone marrow at 8 weeks post bone marrow transfer (Supplementary Fig. 6a).

### Confocal immunofluorescence and analysis
Tumor-bearing mice were anesthetized and transcardially perfused with ice-cold PBS followed by 2.5% Formalin. Brains were removed and post-fixed in 2.5% Formalin at 4 °C overnight and then dehydrated in 30% sucrose at 4 °C for 48 h. Brains were embedded in Tissue Freezing Medium (TFM, General Data Company) and frozen at −80 °C. Frozen brains were made into 25 μm sections using a cryostat and mounted to slides. After outlining sections with a hydrophobic barrier pen, they were washed with TBS + 0.05% Tween-20. Following the initial wash, they were washed with 1% SDS in PBS. After the SDS wash, sections were blocked in TBS + 0.05% Tween-20 + 10% normal donkey serum for 1 h at room temperature. Sections were then incubated with primary antibodies (diluted in TBS + 0.05% Tween-20 + 10% normal donkey serum) (Supplementary Table 2) overnight at 4 °C. The following day, tissue sections were incubated with secondary antibodies (diluted in TBS + 0.05% Tween-20 + 10% normal donkey serum) for 2 h at room temperature. SlowFade Glass (Thermo Fisher Scientific) was then applied before adding a coverslip to the slide. Tumor-bearing hemispheres were imaged on a Dragonfly spinning disc confocal (Andor). Imaris v.9.6.0 (Oxford Instruments) was used to process and analyze images. Tumors were segmented using the surface tool to allow identification of both intra- and peri-tumoral cells. Cells were quantified using the spots tool in Imaris. The percent intratumoral cells is the number of intratumoral cells divided by total cells of that type. The number of cells per mm³ tumor is calculated by dividing the total number of intratumoral cells by the volume of the surface created around the tumor.

### Statistics and reproducibility
Graphs represent the mean ± SEM and are representative of two experimental repeats unless stated otherwise. Statistical tests were completed using GraphPad v.9.2.0 (Prism). When making multiple comparisons in the same graph, a repeated measures two-way ANOVA was performed, and if the interaction term was significant, then a post hoc unpaired two-tailed $t$-test was performed. If only making one comparison in a graph, an unpaired two-tailed Student's $t$-test was used. Asterisks indicate a level of significance (*$p < 0.05$, **$p \leq 0.01$, ***$p < 0.001$, $p > 0.05$ not significant). No statistical methods were used to predetermine sample size.

Survival curves were analyzed using a log-rank (Mantel–Cox) test. Results of independent experiments were combined if the effect of replication was not a significant source of variance in an ANOVA. Mice that underwent treatment were randomized, within their genotype, to treatment groups after tumor injection. All survival studies were monitored with the help of veterinary staff from the Duke animal facility who were blinded to the studies and reported endpoints accordingly.

### Graphical illustrations
Graphical illustrations displayed in Figs. 1i, j, 2a, 4h, 7k, and 8a were created with BioRender.com and exported under a paid subscription.

### Reporting summary
Further information on research design is available in the Nature Research Reporting Summary linked to this article.

## Data availability

Unprocessed scRNA-seq data have been uploaded to NCBI Gene Expression Omnibus under data repository accession number GSE197879. The processed Seurat objects have also been made available through Zenodo under record number https://zenodo.org/record/6654420. The data necessary to reproduce the graphs presented within this manuscript are provided in the Source Data file. Gene markers identified through differential expression conserved between genotypes are available in Supplementary Data 1 and 3. Differentially expressed genes found between genotypes are available in Supplementary Data 2 and 4. Differentially expressed genes found between DC-like and Apoe TAMs are available in Supplementary Data 5. Gene signatures used throughout the manuscript are available in Supplementary Data 7. Additional survival statistics for all survival curves presented are available in Supplementary Data 8. The publicly available data used in this study were obtained from the GENT2 database http://gent2.appex.kr/gent2/, from the GBMseq portal http://gbmseq.org/, and from the Allen Brain Map https://portal.brain-map.org/atlases-and-data/rnaseq/human-m1-10x. The mm10 reference assembly is available through GenBank under accession code GCA_000001635.2. The remaining data are available within the article, Supplementary Information or Source data file. Source Data are provided with this paper.

## Code availability

The R code that was used to perform the scRNA-seq analysis can be found on GitHub: https://github.com/wht10/CT2A_scRNAseq_CaMKK2KOvWT.

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

## Acknowledgements

We would like to acknowledge all the members of the Duke Brain Tumor Immunotherapy Program for insightful comments and advice. This work was supported by NIH Grants: 5R01-NS099463-04, 5U01-NS090284-05, 5P01-CA225622-03, 1R01-CA235612-02, UFDSP00012036, 5P50-CA190991-07, 3P50-CA190991-07S1, and 1R01-NS116888-0A1 awarded to J.H.S. Flow cytometry was performed in the Duke Human Vaccine Institute Flow Cytometry Facility (Durham, NC). Confocal immuno-fluorescence imaging and analysis were performed at the Duke Light Microscopy Core Facility (Durham, NC). 10X scRNA-seq library preparation was performed at the Duke Molecular Physiology Institute (Durham, NC). Sequencing was performed at the Duke Center for Genomic and Computational Biology (Durham, NC).

## Author contributions

W.H.T. conceived the project. W.H.T., L.S.-P., and M.D.G. designed the experiments. W.H.T. wrote the manuscript. All authors (including D.P.M., M.K., D.M.A., and P.E.F.) revised the manuscript. W.H.T., J.W.P., A.M, and M.C. performed flow cytometry experiments. W.H.T. and J.W.P. analyzed flow cytometry data. W.H.T., J.P., J.W.-P., and J.R. performed or helped with bioinformatics analyses. M.C. performed and analyzed immuno-fluorescence experiments. L.R. consulted on the project as a CaMKK2 expert and provided CaMKK2-EGFP and CaMKK2[fl/fl] mice. M.D.G. and J.H.S supervised all aspects of the work.

## Competing interests

P.E.F. reports consulting for Monteris Medical. J.H.S. has an equity interest in Istari Oncology, which has licensed intellectual property from Duke related to the use of poliovirus and D2C7 in the treatment of glioblastoma. J.H.S. is an inventor on patents related to the PEP-CMV DC vaccine with tetanus (US 9974848), as well as poliovirus vaccine (US 11406677) in the treatment of glioblastoma. J.H.S. has an equity interest in Annias Immunotherapeutics, which has licensed intellectual property from Duke University Medical Center related to the use of the pepCMV vaccine in the treatment of glioblastoma. M.K. reports receiving institutional research funding from AbbVie, Bristol Myers Squibb, Celldex and Specialized Therapeutics. M.K. additionally reports consulting/advisory roles with Voyager Therapeutics, AbbVie, Bristol Myers Squibb, Janssen and Janssen, Eli Lilly, Ipsen, Pfizer, and Roche. L.R. and D.P.M. have applied for a patent covering the use of CaMKK2. The remaining authors declare no other competing interests.
