## [Peer Review File · Nature Communications]

Neuronal CaMKK2 promotes immunosuppression and checkpoint blockade resistance in GlioblastomaREVIEWER COMMENTS

Reviewer #1 (Remarks to the Author): with expertise in myeloid cell immunology

Tomaszewski et al report a role for CaMKK2, both in myeloid cells and in neurons, in the promotion of glioblastoma progression and the skewing towards a less immune-permissive TME. scRNA-seq was used to unravel the impact of CaMKK2-deficiency on the immune TME. A role for CaMKK2 in TAM had been suggested before in the context of breast cancer, but the situation in GBM appears to be somewhat different in the sense that CaMKK2 within non-hematopoietic cells, in particular neurons, contributes to tumor progression. Several remarks remain:

- 1) The gating in FigS1c is not entirely correct. cDC1 are almost absent, which is expected when a pre-gating occurs on CD11b-high cells. cDC1 are likely within the 'microglia' gate.
- 2) Information should be given about the size of the tumors that are being compared. Comparing bigger (WT) to smaller (KO) tumors already induces a bias, since bigger and smaller tumors can be expected to show differences, irrespective of the genetic make-up of the host. Some of the observed differences should be checked in similarly sized WT vs KO tumors.
- 3) Why were the TAMs not reclustered, similar to what is done for the T cells? Most likely, additional TAM subsets, beyond Apoe+ and DC-like, will appear, so this should be done.
- 4) Figure 4 entirely deals with an in silico prediction, without biological validation. Cell-cell interactions predicted by algorithms should be considered as hypotheses, not as facts. Biological validation of some of these predicted interactions is needed. Why do the authors highlight CXCL, IFNg, CD40L, MHC-II, but not IL-10, LAIR1, CD200 which are myeloid cell inhibitory ligands/receptors? This seems a very biased choice.
- 5) 'CD4s'. That is not a correct scientific term. I guess the researchers mean 'CD4+ T cells'?
- 6) Since there are more CD4+ T cells in the TME of KO mice, their chance of interacting with myeloid cells increases. Is this a stochastic phenomenon or are the CD4+ T cells better equipped to interact with myeloid cells in the KO situation?
- 7) How is the ICB responsiveness in neuron-specific conditional KO?

Reviewer #2 (Remarks to the Author): with expertise in glioblastoma, tumor immunology

Tomaszewski et al. propose that "stromal CaMKK2 promotes immunosuppression and checkpoint blockade resistance in glioblastoma". It is an interesting and original paper that proposes a role for CaMKK2 in determining the nature of the GBM TME.

The authors may wish to revise the title of the manuscript as they propose also a role for neuronal CaMKK2, which is not included in their title.

Please provide information on the genetic background of all cell lines used, and all transgenic lines used. Are all the different cell lines syngeneic in all the transgenic lines used?

The x-axis of graphs depicting survival should start at 0, not at any other value. Please add median survival and its error to all the survival curves shown throughout the manuscript.

If tumor growth is affected in the CaMKK2-TME is there an effect on tumor cell proliferation? BRDU labeling experiments would provide a clear answer.

Some differences in the survival curves are large (Fig. 1f), while others are only 6-7 days. Please comment on the significance of the smaller changes.

It is difficult to see anything in Fig. S1. Please increase brightness. This problem is also evident in other figures.

The human data in Fig. S1e are very important and could be moved to the main section of the manuscript. Especially, if the authors were to have access to human tumors, a description of CaMKK2 distribution (scRNAseq) therein would be of special importance.

On page 16, Discussion, please temper the comment on monotherapy, line 366, as further below the authors seem to propose combination therapy.

The authors do not propose a particular mechanism of action of CaMKK2. They should be encouraged to do so. The very nice summary figure in Fig. S6c could be moved to the main

manuscript, with the addition of some speculation on the mechanism of action of CaMKK2 (beyond the description of Fig. S6c.

In Fig. 3c, add the color code of the violin plots.

In Fig. 5, a, f, are barely visible.

Equally, in Fig. S4, panel j is barely visible. Please adjust.

In Fig. S5 no statistical significance values is provided in panel 'a'. Please add. Panel b is barely visible.

Figure S6 facilitates the interpretation of the results and could be included with the main figures.

Reviewer #3 (Remarks to the Author): with expertise in glioblastoma, tumor immunology, scRNAseq

The study explored a role of Calmodulin Dependent Kinase Kinase 2 (CaMKK2) in GBM pathology and response to inhibitors of checkpoint blockade (ICB). The rationale was based on the facts of CaMKK2 expression in neurons, glioma cells and glioma associated macrophage (GAMs) and a role of the kinase in the neuronal BDNF expression and its role in neuronal activity dependent glioma growth. CaMKK2 was overexpressed in GBM and its expression negatively correlated with survival. Data from other tumors suggested a role of CaMKK2 in tumor progression.

The authors demonstrate CaMKK2 is most highly expressed in GAMs and neurons in naïve and tumor-bearing using CaMKK2-EGFP reporter mice. They showed high expression of CaMKK2 in pro-tumor cells and association with worsened survival in patients with GBM. The used publicly available scRNA-seq data to confirm that this pattern of expression is similar in the naïve human brain and patient GBM samples. They used CaMKK2^{-/-} (CaMKK2 KO) mice and three GBM tumor models: CT2a, GL261, and Kluc and found that CaMKK2 KO survived significantly longer than controls. Host CaMKK2 was found to reduce survival and promote ICB resistance. The analyses of the TME showed that CaMKK2 is associated with several ICB resistance-associated immune phenotypes. CaMKK2 promoted exhaustion in CD8⁺ T cells and reduced the expansion of effector CD4⁺ cells, reducing their tumor penetrance. CaMKK2 also maintained myeloid cells in a disease-associated microglia-like phenotype. They shows that neuronal CaMKK2 is required for maintaining the myeloid phenotypes associated with ICB resistance and is deleterious to survival. The experiments with bone marrow chimeras identified that CaMKK2 deficiency in non-hematopoietic cells is necessary for ICB response, which indeed is a novel role for CaMKK2 in non-hematopoietic cells as drivers of ICB resistance in GBM. The authors conclude that CaMKK2 as a novel contributor to ICB resistance and CaMKK2 in neurons are drivers of pro-tumor immune phenotypes in GBM. The results are very interesting and have potential clinical implications. Altogether, this study brings a lot of novel and original information despite minor weaknesses and drawbacks of the scRNA seq analysis. By combination of multiple approaches the authors well documented their points and provided a novel information on mechanism ICB resistance in GBM. The work is of great significance to the field.

The authors explored different relevant datasets (TCGA, CCGA, GENR2, public sc-omics data) using standard, appropriate methods. While most of the models and methods used are appropriate, some models were not adequate to provide expected answers (i.e. a lack of myeloid CaMKK2 KO, subcutaneous GBM cell implantation). The methods are not well described and the experiments would be hard to reproduce. The indicated weakness can be improved by a proper revision.

Major comments:

1. Several bulk RNAseq and sc-omics studies showed that GAMs - glioma associated microglia and macrophages - are distinct from peripheral TAMs (tumor associated macrophages), therefore they deserve own name to underlie their uniqueness and the contribution of microglia that have distinct transcriptomic profile than macrophages. I believe using a name GAMs would be more precise.
2. CaMKK2^{-/-} (CaMKK2 KO) mice were used with three GBM tumor models and tumor implanted CaMKK2 KO mice survived significantly longer than controls. The knockout were not myeloid cell specific, therefore, it is hard to pin point a mechanism underlying this effect, whether it was a lack of BDNF from neurons or defective CaMKK2 signaling in GAMs. This was partially overcome by BM transplantation experiment and I appreciate that these results provide a partial answers.
3. They tested a combination therapy consisting of ICB antibodies against PD1 and T-cell

immunoglobulin and mucin-domain containing-3 (TIM3) in subcutaneous (SQ) and intracranial (IC) GBM models in WT mice. I wonder what was a rationale for using subcutaneous tumor models in the work so focused on TME. It is now widely accepted that those are not relevant GBM models due to the lack of appropriate TME. I think these results are not relevant for GBM and any ICB resistance issue as this milieu lacks neurons, astrocytes, microglia and TAMs, most cells contributing to TME and ICB resistance. This is highlighted by the obtained results showing that the combined therapy induced tumor regression in the SQ model, but had no effect on the survival in the IC model.

4. The results of scRNAseq are not fully consistent. Immunophenotyping of the TME showed that numbers of tumor-infiltrating lymphoid and myeloid cells, were not altered in CaMKK2 KO, so how ICB treatment works stronger and in immune-mediated manner?

5. scRNAseq analysis. Were the tumor bearing mice transfused with PBS before tissue processing? In general, the proper steps of preprocessing were applied. However, the description of experiments and data is too vague to evaluate its quality. I am missing some information: a number of living cells in the experiment, resolution during cluster definition, saturation of sequencing, number of genes/cell. A number of analyzed cells is relatively low: 7,000 in total with 1,750 per condition. According to the provided information "11,784 Live CD45+ cells passed QC and the SingleCellExperiment object". This is not an impressive number considering a number of animals and conditions. I am puzzled by the lack of homeostatic microglia in data.

6. They performed scRNA-seq on CD45+ immune cells isolated from CT2A tumors in WT and CaMKK2 KO mice. Cell clusters were annotated using previously published gene expression signatures. First, why CT2A gliomas were used? Secondly, the authors distinguished macrophage clusters (Mertk, Adgre1, Fcgr1) that were grouped as "labeled Nos2+ TAM (Nos2 and Arg1), DC-like TAM (H2-Aa and CD74), and Apoe+ TAM (Apoe 138 and Mrc1)". Were there clusters of microglia and monocytes?

7. The choice of CT2A cells for further analyses is not explained. Immunotherapy with autologous tumor lysates of GL261 and CT2A showed a differential response with better efficacy in CT2A tumors and CT2A are considered to be an immunologically silent tumor (Khalsa, J.K., Cheng, N., Keegan, J. et al. Immune phenotyping of diverse syngeneic murine brain tumors identifies immunologically distinct types. *Nat Commun* 11, 3912 (2020). <https://doi.org/10.1038/s41467-020-17704-5>).

8. For bone marrow chimeras there is no information if irradiation was with head protection or whole body irradiation. Statistical analyses were performed generally according to standards in the field. However, it is stated that "Graphs represent the mean \pm S.E.M. and are representative of two experimental repeats unless stated otherwise". It is accepted to have 3 experimental repeats.

9. Data Availability. Data will be uploaded to NCBI Gene Expression Omnibus (<https://www.ncbi.nlm.nih.gov/geo/>) upon acceptance. I think the newly generated data should be uploaded to the repository before submission and made available for reviewers upon request to give a chance to evaluate their quality.

10. The analysis of human brain scRNAseq data shows CAMKK2 mostly in neurons, very little in microglia. In scRNAseq analysis I noted some inconsistencies in the data. Fig. 2 shows UMAP of CD45 cells and there are microglia but I do not see a population of activated microglia in addition to homeostatic microglia. I do not see proliferating cells which have been reported in murine glioma models. There is no Tregs population, however in Fig. 3 when T cells are analyzed this population is clearly visible. The authors used signatures from human studies (coming mostly from CYTOF data) to identify macrophage subpopulations, instead of signatures used in other scRNAseq mouse studies and it is not clear what Nos2-TAMs mean (Apoe+TAMs likely represent phagocytic cells). None of the populations is marked as immunosuppressive TAMs. Stem-like TILs are enigmatic. Cell to cell interactions analysis (Fig.4d) does not show the enhancement of specific interactions. The verification of Iba1+TILs interactions is shown on confocal images but it is more co-occurrence than interaction demonstration (Z-stack at high magnification should be demonstrated to prove this point).

11. "In summary, we found that CaMKK2 deficiency dramatically remodels the immune TME to a more anti-tumor, ICB-responsive phenotype, and away from phenotypes associated with ICB resistance". This general conclusion is disputable as CaMKK2 deficiency (mostly in neurons) may result in smaller tumors and in turn could lead to smaller immunosuppression in the TME allowing stronger ICB response and lesser ICB resistance. I have my doubt if general CaMKK2 KO could provide conclusive answers regarding its effects on the TME. The authors raised this issue in the discussion and attempted to solve it in the experiments in which they show that CaMKK2 deficiency in

non-hematopoietic cells (CaMKK2 KO in irradiated recipients) was necessary for any survival benefit. Increased survival could be achieved by combination of CaMKK2 deficiency in both the hematopoietic and non-hematopoietic compartments.

Minor comments:

- Please write gene names in italics (i.e. "CaMKK2 expression is associated with a worse prognosis in patients with GBM using The Cancer Genome Atlas (TCGA) and the Chinese Glioma Genome Atlas (CGGA)").
- In the first figure cells are defined as Kluc but in the methods the name is KR158B-Luc, which is slightly confusing and should be unified.

Dear Referees,

Thank you for your thoughtful review of our manuscript, which is currently entitled, “**Stromal CaMKK2 promotes immunosuppression and checkpoint blockade resistance in Glioblastoma**”, although as per Referee #2’s first comment, we would like to change the title to, “**Neuronal CaMKK2 promotes immunosuppression and checkpoint blockade resistance in Glioblastoma**”. Below, please find a point-by-point response to the italicized comments from the Referees. The attached revision of the manuscript details the changes made in red text and highlights areas in Comment Boxes where we provide specific responses to the concerns of the Referees.

Summary of Figure Changes

Figure 1 – Survival curves (Fig. 1d-h) were adjusted to all start at 0.

Supplementary Figure 1 – Image quality was improved for Fig. S1a.

Figure 3 – Legend was added for Fig. 3c.

Figure 5 – Size of points in Fig 5a were enlarged to improve visibility. Survival curve in Fig. 5e was adjusted to start at 0. Image quality of Fig. 5f was improved.

Supplementary Figure 4 – Image quality of Fig. S4j was improved. Fig. S4k includes a single z-plane of the tumor microenvironment at high magnification to identify CD4+ Iba1+ cell interactions.

Supplementary Figure 5 – Indications of significance were added to Fig. S5a. Image quality was improved for Fig. S5b.

Figure 7 – Survival curves in Fig. 7a,c,d,i were modified to start at 0. New data was added in Fig. 7j to demonstrate that neuronal CaMKK2 deficiency is sufficient to elicit ICB response. Fig. 7k

was moved from Fig. S6 to provide a visual aid for understanding the chimera generation process. The title of Fig. 7 was modified to reflect the findings in Fig. 7j.

Supplementary Figure 6 – Fig. S6b,c were added to expand upon CaMKK2's mechanism as they demonstrate that hematopoietic CaMKK2 deficiency is sufficient to phenocopy T cell phenotypes observed in total CaMKK2 deficiency.

Figure 8 – The graphical summary was moved from Fig. S6 to a main figure as per the request of Referee 2 comment # 9.

Response to Referees

Detailed responses to Referees are outlined below:

Referee 1

General Remarks: *Tomaszewski et al report a role for CaMKK2, both in myeloid cells and in neurons, in the promotion of glioblastoma progression and the skewing towards a less immune-permissive TME. scRNA-seq was used to unravel the impact of CaMKK2-deficiency on the immune TME. A role for CaMKK2 in TAM had been suggested before in the context of breast cancer, but the situation in GBM appears to be somewhat different in the sense that CaMKK2 within non-hematopoietic cells, in particular neurons, contributes to tumor progression. Several remarks remain:*

Comment 1: *The gating in FigS1c is not entirely correct. cDC1 are almost absent, which is expected when a pre-gating occurs on CD11b-high cells. cDC1 are likely within the 'microglia' gate.*

Response: We thank the reviewer for bringing attention to this. In Fig. 2c, we demonstrate that cDC1s have a minimal presence in the tumor microenvironment using scRNA-seq, which is an unbiased approach. To examine if cDC1 cells are within the microglia gating we further examined the microglia population below. The data below demonstrates an absence of cDC1 contamination within the Microglia gate. We believe that the CD11c-int MHCII-int gate here represents a population of activated microglia and not contaminating cDCs, as microglia are known to upregulate MHCII and CD11c upon activation (<https://doi.org/10.3389/fimmu.2015.00249>).

Comment 2: Information should be given about the size of the tumors that are being compared. Comparing bigger (WT) to smaller (KO) tumors already induces a bias, since bigger and smaller tumors can be expected to show differences, irrespective of the genetic make-up of the host. Some of the observed differences should be checked in similarly sized WT vs KO tumors.

Response: We appreciate the importance of disentangling immunophenotypic changes that may be driven indirectly by CaMKK2 via reductions in tumor size versus the direct effect of CaMKK2 acting in a cell-intrinsic manner. Fig. 7b,g,h demonstrate that restriction of the MHCII-high macrophage phenotype is not driven by cell-intrinsic CaMKK2 expression and is instead an indirect effect of CaMKK2 in neurons.

Considering that the MHCII-high TAM phenotype observed in the CaMKK2 deficient neuron model may be an indirect effect of smaller tumors, we sought to determine what effect tumor size had in the absence of neuronal CaMKK2 on TAM phenotype (data shown below). We found that an approximate tripling of tumor size did not have a significant effect on TAM phenotype, suggesting that tumor size is not the primary driver of TAM phenotype.

Fig. 7c shows that there are pro-tumor effects of CaMKK2 in the hematopoietic system however, as evidenced by the increase in survival in the CaMKK2 KO Bone Marrow -> CaMKK2 KO Host group relative to the WT Bone Marrow -> CaMKK2 KO Host group. In Fig. S6b,c we have added additional data (in response to Referee 2 comment # 9) showing that having CaMKK2 KO Bone Marrow alone is sufficient to phenocopy the

reduced CD8 exhaustion seen when both host and bone marrow CaMKK2 is absent, which suggests that CaMKK2 in the hematopoietic system is driving CD8 T cell exhaustion, perhaps in a cell-intrinsic manner.

Comment 3: *Why were the TAMs not reclustered, similar to what is done for the T cells? Most likely, additional TAM subsets, beyond Apoe+ and DC-like, will appear, so this should be done.*

Response: Thank you for this comment regarding the myeloid heterogeneity within our scRNA-seq data. Our stated purpose for utilizing scRNA-seq was to resolve genotype-dependent phenotypic differences within major immune cell types. T cells were reclustered because no phenotypically distinct subsets were detected during the initial clustering (Fig. 2b). Upon reclustering of the T cells, genotype-specific clusters like Effector CD4s became apparent (Fig. 5a). In the myeloid population, our initial clustering in Fig. 2b did however reveal genotype-specific clusters within the mononuclear phagocyte (MNP) system such as Apoe+ and DC-like TAMs. As per your request, we have further reclustered the MNPs using the same process we used to recluster the T cells to examine any further heterogeneity within this population (data shown below). Our original clustering identified 6 clusters within the MNPs, and reclustering identified 9. Clusters 8, 7, and 4 remain unchanged. Cluster 3 represents wildtype classical monocytes (cMonos), and cluster 5 represents primarily CaMKK2 KO cMonos. Cluster 0 and 8 are subtypes of DC-like TAMs, and clusters 1 and 2 are subtypes of Apoe+ TAMs. Considering that genotype-dependent differences within the original clusters were discussed in Fig. 6e we feel that the conclusions obtained from the reclustering would not alter our interpretation of the data. We hope the referee agrees with our thought process

of keeping this analysis as data not shown. We do plan to make our source data public to the scientific community for transparency purposes and encourage its further analysis.

Comment 4: Figure 4 entirely deals with an *in silico* prediction, without biological validation.

Cell-cell interactions predicted by algorithms should be considered as hypotheses, not as facts.

Biological validation of some of these predicted interactions is needed.

Why do the authors highlight CXCL, IFN γ , CD40L, MHC-II, but not IL-10, LAIR1, CD200

which are myeloid cell inhibitory ligands/receptors? This seems a very biased choice.

Response: We appreciate the limitations of these *in silico* predictions that you point out.

We have modified the language within the section to provide more clarity when referring

to the data as inferences and predictions. The predicted receptor-ligand interactions that

we highlighted are CXCL9/10 and CXCR3, MHCII and CD4, CD40 and CD40L, and

IFN γ and IFN γ -receptor. The hypothesis we generated from these data was that an MHC-

II+ CD40+ myeloid cell was interacting with a CD4+ CD40L+ IFN γ + cell in the CaMKK2 KO TME. In Fig. 5b we biologically validate the presence of a CD4+ CD40L+ IFN γ + cell by flow cytometry. In Fig. 6h we biologically validate the presence of a MHC-II+ CD40+ myeloid cells by flow cytometry as well. CXCL9 and CXCL10 are T-cell chemoattractants, and it stands to reason that increased secretion of these chemokines should result in increased T-cell accumulation within the TME. In Fig.S4d, we show that there is an increased number of CD4+ T cells by flow cytometry, and in Fig. 5f-h we also validate this by confocal immunofluorescence. In Referee 3's 10th comment they suggest demonstrating interaction between CD4 and myeloid cells using a single z-stack high magnification image showing that these cells are interacting and not merely co-occurring, which we have provided in Fig. S4k, along with the previous data quantifying the increased interactions between myeloid and CD4+ cells in the CaMKK2 KO TME. Together we interpret these data to mean that within the CaMKK2 KO TME there are increased interactions between CD4+ and myeloid cells and within CD4+ T cells there is an increased frequency of CD40L+ IFN γ + expressing cells, and within myeloid cells there is an increased frequency of CD40+ MHCII+ expressing cells, which we interpret as a validation of our hypothesis that was generated in Fig. 4.

We utilized this cell-cell interaction analysis to generate hypotheses that may explain the ICB-responsiveness in the CaMKK2 KO mice. As a result, we were primarily looking for immunostimulatory interactions as opposed to inhibitory, as they are better suited to explain the ICB-responsiveness. Fig. S3a summarizes the cell-types that are active in the receptor-ligand pathways. In the case of IL10, you can see that the main outgoing cell-type that is expressing IL10 is CD4+ T cells (likely regulatory T cells), and the main difference in incoming signaling between wildtype and CaMKK2 KO are pDCs, which

are minor components of the TME (Fig. 2c). This is a similar case for LAIR1, which has a higher incoming signaling pattern in DC2 and DC3 than in wildtype, which are also minor components of the TME. The CD200 pathway is primarily an interaction between DC3s and basophils, which we don't expect to have a profound impact on ICB-response due to their limited frequency. Instead, we chose to highlight pathways that were active between T-cells and macrophages that had the potential for immunostimulatory effect and may explain the ICB-response observed in CaMKK2 KO mice.

Comment 5: *'CD4s'. That is not a correct scientific term. I guess the researchers mean 'CD4+ T cells'?*

Response: Thank you for bringing this to our attention. Your assumption is correct, and the instances of 'CD4s' have been changed to 'CD4⁺ T Cells' or 'CD4⁺ TILs (Tumor-infiltrating Lymphocytes)'.

Comment 6: *Since there are more CD4+ T cells in the TME of KO mice, their chance of interacting with myeloid cells increases. Is this a stochastic phenomenon or are the CD4+ T cells better equipped to interact with myeloid cells in the KO situation?*

Response: Due to the increased expression of MHCII and CD40 on myeloid cells (Fig. 6h), and the increased expression of CD40L on CD4⁺ T cells (Fig. 5c), we believe that these cells are better equipped to interact with each other in the CaMKK2 KO TME.

Comment 7: *How is the ICB responsiveness in neuron-specific conditional KO?*

Response: Thank you for suggesting this experiment. We did not perform this experiment originally due to the limited availability of the conditional knockout mice. This experiment has demonstrated that CaMKK2 in neurons is a significant contributor to ICB resistance. We have now included this data in Fig. 7j (and below), and have added the following sentence to the results and discussion sections.

Furthermore, deletion of CaMKK2 in neurons was also sufficient to improve survival and ICB response (Fig. 7i,j). This implicates neuronal CaMKK2 as a key contributor to the tumor-promoting TAM phenotype, tumor progression, and ICB resistance within the GBM TME.

Using this approach, we identified that CaMKK2 deficiency in neurons is sufficient to

induce ICB response and anti-tumor immune phenotypes.

This led to our identification of neurons as important in maintaining an MHCII^{low} TAM phenotype, promoting tumor progression, and stimulating ICB resistance.

Additionally, we have modified the final sentences of the abstract to read as follows.

Lastly, neuronal CaMKK2 is required for maintaining the ICB resistance-associated myeloid phenotype, is deleterious to survival, and promotes ICB resistance. Our findings reveal CaMKK2 as a novel contributor to ICB resistance and newly identify neurons as a driver of immunotherapeutic resistance in GBM.

Referee 2

General Remarks: *Tomaszewski et al. propose that "stromal CaMKK2 promotes immunosuppression and checkpoint blockade resistance in glioblastoma". It is an interesting and original paper that proposes a role for CaMKK2 in determining the nature of the GBM TME.*

Comment 1: *The authors may wish to revise the title of the manuscript as they propose also a role for neuronal CaMKK2, which is not included in their title.*

Response: Thank you for this insight. Considering the results discussed in the response to Referee 1 comment # 7, we would like to change the title to, “Neuronal CaMKK2 promotes immunosuppression and checkpoint blockade resistance in Glioblastoma”.

Comment 2: *Please provide information on the genetic background of all cell lines used, and all transgenic lines used. Are all the different cell lines syngeneic in all the transgenic lines used?*

Response: We appreciate the clarification that is necessary here. The background of all cell lines used were derived from mice with the C57BL/6 background, and all the transgenic mouse lines were derived from or backcrossed to the C57BL/6 background.

These tumor models are thus syngenic to all the transgenic mouse models. The below sentence was added under the ‘**Mice**’ heading in the methods section.

All transgenic mouse lines were derived from or have been previously backcrossed to the C57BL/6 background.

Comment 3: *The x-axis of graphs depicting survival should start at 0, not at any other valued.*

Please add median survival and its error to all the survival curves shown throughout the manuscript.

Response: We appreciate this comment. We have modified all the survival graphs so they start at 0 and will provide a supplementary table detailing the median survival and its error for all the survival data.

Comment 4: *If tumor growth is affected in the CaMKK2-TME is there an effect on tumor cell proliferation? BRDU labeling experiments would provide a clear answer.*

Response: Thank you for this comment. We believe that if there is an effect on tumor cell proliferation resulting from CaMMK2 deficiency in the TME its effect on survival and tumor growth is likely overshadowed by the induction of tumor cytotoxic T cell immune response. In our CD4+ and CD8+ T cell depletion experiments, we found that increased survival in CaMKK2 deficient mice was dependent on these T cells. We additionally found that these T cells were more pro-inflammatory and cytotoxic, relative to wildtype, in subsequent phenotyping. This demonstrates that tumor clearance and improved survival in the CaMKK2 deficient mice are dependent on cytotoxic T cells

specifically, and the immune system more generally. While this doesn't exclude the possibility that the CaMKK2 deficient host alters tumor proliferation, it does indicate that cytolytic function is the primary driver of improved survival and tumor clearance.

In response to your 9th comment regarding further elucidating the mechanism of action for CaMKK2, we provide new data in Fig. S6c indicating that hematopoietic CaMKK2 deficiency is sufficient to phenocopy the reduced exhaustion observed in total body CaMKK2 KO mice. This suggests that the improved cytotoxic capacity of TILs results from CaMKK2 deficiency within the immune system and is unlikely to be a downstream effect of smaller or less proliferative tumors.

In response to Referee 1 comment # 2, we additionally show that tumor growth/size does not significantly affect macrophage phenotype, suggesting that this phenotype is also agnostic to tumor size.

Comment 5: Some differences in the survival curves are large (Fig. 1f), while others are only 6-7 days. Please comment on the significance of the smaller changes.

Response: We appreciate that there is certainly variability in the outcomes of these survival studies, despite our best efforts to reproduce the experimental conditions as closely as possible between experiments and replicates. In general, it seems that there may be isotype and radiation effects that reduce the observed survival differences. To bring this to the reader's attention, we have added the below sentences to the discussion.

It is worth noting here that we observed that the chimera generation process, as well as isotype antibody treatment, had negative effects on survival, which generally led to smaller - although still significant - differences in survival compared to untreated survival studies. In the CT2a preclinical model, the combination of CaMKK2 deficiency along with ICB treatment approximately doubles median survival which we believe is suggestive that CaMKK2 may be a clinically significant target in GBM.

Comment 6: *It is difficult to see anything in Fig. S1. Please increase brightness. This problem is also evident in other figures.*

Response: Thank you for notifying us about the lack of clarity in this image. We have improved the visibility of these images.

Comment 7: *The human data in Fig. S1e are very important and could be moved to the main section of the manuscript. Especially, if the authors were to have access to human tumors, a description of CaMKK2 distribution (scRNAseq) therein would be of special importance.*

Response: We agree that human sequencing data carries special importance, however we did not acquire novel scRNA-seq data on human tumors. We did however reanalyze publicly available human naïve and tumor-bearing brain data shown in **Fig. 1b** and **Fig. S1e**. Due to space constraints, we thought it prudent to include the publicly available human glioma data in the main figure and include the publicly available naïve brain data in the supplement. Thematically we feel this is more consistent so that the CaMKK2 expression in tumor-bearing conditions (murine and human) is in the main figure, and naïve CaMKK2 expression is in the supplement.

Comment 8: *On page 16, Discussion, please temper the comment on monotherapy, line 366, as further below the authors seem to propose combination therapy.*

Response: Thank you for bringing attention to the tone of the wording here. We have changed the wording regarding the potential efficacy of monotherapy to read as follows:

Because CaMKK2 appears to have pro-tumor functions in human GBM, and deletion of CaMKK2 extends survival in preclinical models, we expect that a brain penetrant CaMKK2 inhibitor may be efficacious as a monotherapy.

Comment 9: *The authors do not propose a particular mechanism of action of CaMKK2. They should be encouraged to do so. The very nice summary figure in Fig. S6c could be moved to the main manuscript, with the addition of some speculation on the mechanism of action of CaMKK2 (beyond the description of Fig. S6c.*

Response: Thank you for this suggestion. In order to further elucidate CaMKK2's mechanism of action, we have included additional experiments in Fig. S6b,c. (also shown below) We have additionally moved the summary figure to its' own main figure and discussed it in more depth in the discussion section. You can find the text edits accompanying these changes below.

Additional chimera experiments did however indicate that non-hematopoietic CaMKK2 deficiency was dispensable for the CD4+ TIL skewing, and reduced CD8+ TIL exhaustion shown in Fig. S4d and Fig. 3g (Fig. S6b,c). This indicates that hematopoietic CaMKK2 deficiency is sufficient to promote these anti-tumor TIL phenotypes.

Collectively, these observations led us to construct our working model (Fig. 8). Neuronal CaMKK2 has profound pro-tumor effects, demonstrated by its ability to maintain TAMs in an ICB-resistance associated phenotype, as well as promote tumor progression and ICB

resistance. CaMKK2 within the hematopoietic additionally has pro-tumor effects as evidenced by the bone marrow chimera experiments. Fig. 7c indicates that hematopoietic CaMKK2 deficiency extends survival in the context of non-hematopoietic CaMKK2 deficiency. Additionally, Fig. S6b,c suggests that CaMKK2 in the hematopoietic compartment is sufficient to phenocopy TIL phenotypes that were observed in the germline CaMKK2 KO mice. We hypothesize that neuronal CaMKK2 is a primary driver of ICB-resistance via maintaining TAMs, an abundant component of the GBM TME, in a pro-tumor phenotype. The pro-tumor TIL phenotypes seem to be driven by hematopoietic CaMKK2 expression, likely as a combination of expression in innate and lymphoid cells. As a result, CaMKK2 promotes tumor progression, ICB-resistance, and pro-tumor immune phenotypes.

Comment 10: *In Fig. 3c, add the color code of the violin plots.*

Response: Thank you for bringing attention to this missing detail. We have corrected it in the revised figure.

Comment 11: *In Fig. 5, a, f, are barely visible.*

Response: Thank you for bringing this to our attention. We have improved the visibility of these figures.

Comment 12: *Equally, in Fig. S4, panel j is barely visible. Please adjust.*

Response: Thank you for bringing this to your attention. We have improved the visibility of these images.

Comment 13: *In Fig. S5 no statistical significance values is provided in panel 'a'. Please add. Panel b is barely visible.*

Response: Thank you for notifying us about this. We have included statistical comparisons for Fig. S5a and have improved the image quality in of S5b.

Comment 14: *Figure S6 facilitates the interpretation of the results and could be included with the main figures.*

Response: Thank you for this note. We have included these figures at the end of Fig. 7 to better facilitate the interpretation of the chimera experiments.

Referee 3

General Remarks: *The study explored a role of Calmodulin Dependent Kinase Kinase 2 (CaMKK2) in GBM pathology and response to inhibitors of checkpoint blockade (ICB). The rationale was based on the facts of CAMKK2 expression in neurons, glioma cells and glioma associated macrophage (GAMs) and a role of the kinase in the neuronal BDNF expression and its role in neuronal activity dependent glioma growth. CaMKK2 was overexpressed in GBM and its expression negatively correlated with survival. Data from other tumors suggested a role of CaMKK2 in tumor progression.*

The authors demonstrate CaMKK2 is most highly expressed in GAMs and neurons in naïve and tumor-bearing using CaMKK2-EGFP reporter mice. They showed high expression of CaMKK2 in pro-tumor cells and association with worsened survival in patients with GBM. They used publicly available scRNA-seq data to confirm that this pattern of expression is similar in the naïve human brain and patient GBM samples. They used CaMKK2^{-/-} (CaMKK2 KO) mice and three GBM tumor models: CT2a, GL261, and Kluc and found that CaMKK2 KO survived significantly longer than controls. Host CaMKK2 was found to reduce survival and promote ICB resistance. The analyses of the TME showed that CaMKK2 is associated with several ICB resistance-associated immune phenotypes. CaMKK2 promoted exhaustion in CD8⁺ T cells and reduced the expansion of effector CD4⁺ cells, reducing their tumor penetrance. CaMKK2 also maintained myeloid cells in a disease-associated microglia-like phenotype. They shows that neuronal CaMKK2 is required for maintaining the myeloid phenotypes associated with ICB resistance and is deleterious to survival. The experiments with bone marrow chimeras identified that CaMKK2 deficiency in non-hematopoietic cells is necessary for ICB response , which indeed is a novel role for CaMKK2 in non-hematopoietic cells as drivers of ICB resistance in GBM. The authors conclude that CaMKK2 as a novel contributor to ICB resistance and CaMKK2 in neurons are drivers of pro-tumor immune phenotypes in GBM. The results are very interesting and have potential clinical implications. Altogether, this study brings a lot of novel and original information despite minor weaknesses and drawbacks of the scRNA seq analysis. By combination of multiple approaches the authors well documented theirs points and provided a novel information on mechanism ICB resistance in GBM. The work is of great significance to the field.

The authors explored different relevant datasets (TCGA, CCGA, GENR2, public sc-omics data)

using standard, appropriate methods. While most of the models and methods used are appropriate, some models were not adequate to provide expected answers (i.e. a lack of myeloid CaMKK2 KO, subcutaneous GBM cell implantation). The methods are not well described and the experiments would be hard to reproduce. The indicated weakness can be improved by a proper revision.

Major comments:

Major Comment 1: *Several bulk RNAseq and sc-omics studies showed that GAMs - glioma associated microglia and macrophages - are distinct from peripheral TAMs (tumor associated macrophages), therefore they deserve own name to underlie their uniqueness and the contribution of microglia that have distinct transcriptomic profile than macrophages. I believe using a name GAMs would be more precise.*

Response: Thank you for bringing attention to potential confusion regarding how we have termed these myeloid subpopulations. We agree that it is more confusing when microglia are included under the “TAM” umbrella term and that the terminology should be inclusive to those who use the GAM term in the case of glioma. Within the manuscript, we refer to the mononuclear phagocyte (MNP) system, which includes bone marrow-derived macrophages and monocytes as well as yolk-sac derived, tissue-resident microglia. In an effort to keep our terminology more precise we refer to macrophages, microglia, and monocytes directly and separately so as to not conflate them. To clarify this, we have edited the manuscript to include the below adjustments.

The etiology of ICB resistance in GBM remains to be fully elucidated but is thought to be linked to the immunosuppressive nature of the tumor microenvironment (TME) and the pro-tumor function of stromal cells, including tumor-associated macrophages (TAMs), also sometimes referred to as glioma-associated macrophages in GBM.

Given the increase in immunostimulatory programming of the mononuclear phagocyte (MNP) system (bone marrow-derived TAMs, and monocytes as well as yolk-sac derived tissue-resident Microglia), as well as the requirement for T cells in mediating the survival benefit in CaMKK2 KO mice, we further analyzed the TIL compartment.

Major Comment 2: *CaMKK2^{-/-} (CaMKK2 KO) mice were used with three GBM tumor models and tumor implanted CaMKK2 KO mice survived significantly longer than controls. The knockout were not myeloid cell specific, therefore, it is hard to pin point a mechanism underlying this effect, whether it was a lack of BDNF from neurons or defective CaMKK2 signaling in GAMs. This was partially overcome by BM transplantation experiment and I appreciate that these results provide a partial answers.*

Response: Thank you for acknowledging this, which stems from issues associated with studying a gene that is nearly ubiquitously and lowly expressed. We have expanded on our understanding of the CaMKK2's mechanism, which leverages conditional knockout models in response to Referee 2's comment # 9. In short, we have demonstrated that neuronal CaMKK2 deficiency is capable of polarizing macrophages to a more immunostimulatory phenotype, prolonging survival, and inducing ICB responsiveness

(Fig. 7g-j). This is in contrast to myeloid CaMKK2 deficiency, which was insufficient to extend survival or polarize macrophages to an immunostimulatory phenotype (Fig. 7a,b).

Major Comment 3: *They tested a combination therapy consisting of ICB antibodies against PD1 and T-cell immunoglobulin and mucin-domain containing-3 (TIM3) in subcutaneous (SQ) and intracranial (IC) GBM models in WT mice. I wonder what was a rationale for using subcutaneous tumor models in the work so focused on TME. It is now widely accepted that those are not relevant GBM models due to the lack of appropriate TME. I think these results are not relevant for GBM and any ICB resistance issue as this milieu lacks neurons, astrocytes, microglia and BAMs, most cells contributing to TME and ICB resistance. This is highlighted by the obtained results showing that the combined therapy induced tumor regression in the SQ model, but had no effect on the survival in the IC model.*

Response: Our rationale for performing the SQ ICB experiment, was to highlight, as you stated, that there is something different about the SQ TME and the IC TME, namely that it lacks numerous brain-resident cell types. We believe the inclusion of this data provides a rationale for further studying the role of brain-resident cell types in promoting ICB resistance. In the manuscript, we go on to show that CaMKK2 in neurons has pro-tumor and immunosuppressive roles. Because neurons are less abundant in the SQ TME than in the IC TME, we think this may partially explain the differences in ICB responsiveness between these tumor locations. Indeed, the experiment requested by Referee 1 in comment # 7 has demonstrated that neuronal CaMKK2 is a significant contributor to ICB resistance.

Major Comment 4: *The results of scRNaseq are not fully consistent. Immunophenotyping of the TME showed that numbers of tumor-infiltrating lymphoid and myeloid cells, were not altered in CaMKK2 KO, so how ICB treatment works stronger and in immune-mediated manner?*

Response: In the manuscript we initially assay the abundance of the major immune subsets without delving into any phenotypic differences within those immune subsets. Due to the higher dimensionality of scRNA-seq data, relative to flow cytometry, it is more capable of identifying phenotypic differences. In Fig. S1g and Fig. S5b, we show that the overall abundance of myeloid cells is not significantly different between genotypes, however in Fig. 6 and Fig. S5b we show that the phenotypic and localization differences in these myeloid cells are significantly different between genotypes. To highlight why scRNA-seq may be more sensitive to phenotypic differences than flow cytometry, we have included the below sentence.

Indeed, due to the higher dimensionality of scRNA-seq data relative to flow cytometry data, phenotypic differences among the major immune subsets became apparent.

With regards to the mechanism of how ICB treatment works stronger and in immune-mediated manner to mediate tumor rejections: Our current understanding is that ICB treatment works in an immune-mediated manner because when we deplete CD8⁺ (Fig. 1h) or CD4⁺ T cells (Fig. 5e) the survival benefit is lost. This indicates that tumor control and extended survival in the CaMKK2 KO mice is dependent on T cells specifically and the immune system by extension. We demonstrate that the ICB response observed in the CaMKK2 KO mice is accompanied by phenotypic (increased CD8⁺ T cell cytotoxicity, reduced CD8 T cell exhaustion, increased expression of stem-like markers on CD8⁺ T

cells, increased effector CD4⁺ T cell abundance, re-education from a DAM-like TAM phenotype to an immunostimulatory TAM phenotype) and localization changes (increased intertumoral CD4 T cells) associated with ICB response within the major immune subsets, and not necessarily major shifts in the abundance of the major immune subsets.

Major Comment 5: scRNAseq analysis. *Were the tumor bearing mice transfused with PBS before tissue processing? In general, the proper steps of preprocessing were applied. However, the description of experiments and data is too vague to evaluate its quality. I am missing some information: a number of living cells in the experiment, resolution during cluster definition, saturation of sequencing, number of genes/cell. A number of analyzed cells is relatively low: 7,000 in total with 1,750 per condition. According to the provides information “ 11,784 Live CD45⁺ cells passed QC and the SingleCellExperiment object”. This is not an impressive number considering a number of animals and conditions. I am puzzled by the lack of homeostatic microglia in data.*

Response: Thank you for identifying these details that will improve reproducibility.

Transcardial PBS perfusion was performed prior to harvest of the tumor-bearing hemisphere. This detail was added to the below sentence in the methods section.

In brief, after transcardial perfusion with PBS tumor-bearing hemispheres were harvested on day 14 post tumor implantation.

Regarding scRNA-seq, after sorting for live CD45⁺ cells 14,000 cells were targeted for barcoding. Because these cells were negative for viability dye, they were almost entirely

viable when barcoded. The WT library was sequenced to 56.2% saturation and a median of 3,233 genes per cell were detected. The CaMKK2^{-/-} library was sequenced to 59.6% saturation and a median of 2,921 genes per cell were detected. A resolution of .5 was used for clustering. These details have been added in the following two sentences to the methods section.

The WT library was sequenced to 56.2% saturation and a median of 3,233 genes per cell were detected. The CaMKK2^{-/-} library was sequenced to 59.6% saturation and a median of 2,921 genes per cell were detected.

Thirty principal components were used for clustering and dimensionality reduction using FindNeighbors, RunUMAP, and a resolution of .5 was used for FindClusters in Seurat.

While we also would have liked to have analyzed more cells, we were somewhat cost-limited. Despite this, we detected strong phenotypic differences within major immune subsets between genotypes and are satisfied with the results we obtained with the number of cells analyzed.

Although there are likely heterogeneous functional states within this cluster, we did identify a cluster of Microglia in our analysis. Microglia here are labeled as “Microglia” (Mustard colored, bottom right, of Fig. 2b). In Fig. 2d and Fig. S2a we see that this Microglia cluster highly expresses canonical lineage defining microglia markers such as *Crybb1*, *Sall1*, *P2ry12*, and *Tmem119*.

Major Comment 6: *They performed scRNA-seq on CD45+ immune cells isolated from CT2A tumors in WT and CaMKKK2 KO mice. Cell clusters were annotated using previously published gene expression signatures. First, why CT2A gliomas were used? Secondly, the authors distinguished macrophage clusters (Mertk, Adgre1, Fcgr1) that were grouped as “labeled Nos2+ TAM (Nos2 and Arg1), DC-like TAM (H2-Aa and CD74), and Apoe+ TAM (Apoe 138 and Mrc1)”. Were are clusters of microglia and monocytes?*

Response: The preclinical CT2a glioma model was used on the basis of its low immunogenicity, the fact that it is astrocytic in origin, and that it does not respond to ICB when implanted intracranially. Conversely, GL261 is more immunogenic, not astrocytic in origin, and can respond to ICB when implanted intracranially. We restricted our consideration of preclinical models to ones that were syngeneic to the C57BL/6 background because we wanted to utilize the plethora of transgenic mice with this background. Because we wanted to study the mechanisms of ICB resistance in glioma and perform the most clinically relevant research possible with orthotopic models, we chose to use CT2a due to how it more closely resembles key features of clinical glioma.

Clusters of Microglia and monocytes can be seen in Fig. 2b, and their genetic signatures can be found in Fig. 2d and Fig. S2a. Microglia here are labeled as “Microglia” (Mustard colored, bottom right, of Fig. 2b) and monocytes are labeled as “cMono” (Avacado-green colored, next to DC-like TAM and Apoe+ TAM, Fig. 2b) or “non-cMono” (Green colored, to the right of DC-like TAM, **Fig. 2b**), which refers to classical monocytes and non-classical monocytes respectively.

Major Comment 7: *The choice of CT2A cells for further analyses is not explained.*

Immunotherapy with autologous tumor lysates of GL261 and CT2A showed a differential response with better efficacy in CT2A tumors and CT2A are considered to be an immunologically silent tumor (Khalsa, J.K., Cheng, N., Keegan, J. et al. Immune phenotyping of diverse syngeneic murine brain tumors identifies immunologically distinct types. Nat Commun 11, 3912 (2020). <https://doi.org/10.1038/s41467-020-17704-5>).

Response: Thank you for bringing our attention to this study. In this study, the authors state, “Although GL261 is widely used in preclinical studies, these tumor-bearing mice have significantly fewer APCs and more T cells than GBM patients that are usually suppressive in nature. This immune-phenotype could be a factor contributing to better efficacy of treatment modalities that target T-cell populations in GL261 in comparison to other mouse models”. Because we were studying the response to ICB, a treatment modality that targets T-cell populations, we chose to not use this model for the bulk of our studies. Because human GBM is more immunologically silent we believe that studying ICB response in CT2a has the most potential for clinical translation. To explain the rationale for utilizing CT2a we have added the below sentence to the manuscript.

CT2a was chosen for this and other experiments due to its low immunogenicity, and histological similarities with human GBM.

Major Comment 8: *For bone marrow chimeras there is no information if irradiation was with head protection or whole body irradiation. Statistical analyses were performed generally according to standards in the field. However, it is stated that „Graphs represent the mean ±*

S.E.M. and are representative of two experimental repeats unless stated otherwise”. It is accepted to have 3 experimental repeats.

Response: Thank you for pointing out that missing detail regarding bone marrow chimera generation. We have revised that section of the description of the method to read as follows.

Recipient mice received whole-body irradiation with a 9Gy dose from a Cesium irradiator (Mark I-68A ¹³⁷Cs irradiator, JL Shepherd and Associates).

We certainly agree that more experimental repeats are generally preferred, but because almost all the experiments in this manuscript are either *in-vivo* or performed on *ex-vivo* tissue, experimental triplicates would have been cost and time prohibitive unless needed to confirm a result in question. We consulted with our biostatistician to discuss the data obtained and the potential risks of type I error and false positivity with two experimental repeats. Our statistical consult indicated the likelihood of receiving a type I error twice in a row at $p = .05$ is .25%, which is already a minimal likelihood of a false positive. Furthermore, many of our results have p-values far smaller than .05, which further reduces the likelihood of a type I error. Additionally, many of our findings are verified by two complementary methods (scRNA-seq and flow cytometry for example). We hope that you will find this satisfactory.

Major Comment 9: Data Availability. *Data will be uploaded to NCBI Gene Expression*

Omnibus <https://www.ncbi.nlm.nih.gov/geo/>) upon acceptance. I think the newly generate data

should be upload to the repository before submission and make available for reviewers upon request to give a chance to evaluate their quality.

Response: Thank you for notifying us about this. You can review the unprocessed data under GEO accession GSE197879 by visiting [https://urldefense.com/v3/https://www.ncbi.nlm.nih.gov/geo/query/acc.cgi?acc=GSE197879;!!OToaGQ!-e0-QAN9AzxzuaXnDISPGNctrJa1Zy6pjjLN0fen5Pq2cSr2FESbhDzQOehlKxU\\$](https://urldefense.com/v3/https://www.ncbi.nlm.nih.gov/geo/query/acc.cgi?acc=GSE197879;!!OToaGQ!-e0-QAN9AzxzuaXnDISPGNctrJa1Zy6pjjLN0fen5Pq2cSr2FESbhDzQOehlKxU$) and entering the token yjwjyimsnnwnxgj into the box. We have additionally made the processed Seurat objects available through zenodo. The revised data availability statement has been changed to the below:

Unprocessed scRNA-seq data has been uploaded to NCBI Gene Expression Omnibus ((<https://www.ncbi.nlm.nih.gov/geo/>) under data repository accession number <https://www.ncbi.nlm.nih.gov/geo/query/acc.cgi?acc=GSE197879>. The processed Seurat objects have also been made available through zenodo under record number <https://zenodo.org/record/6654420>.

Major Comment 10: *The analysis of human brain scRNAseq data shows CAMKK2 mostly in neurons, very little in microglia. In scRNAseq analysis I noted some inconsistencies in the data. Fig. 2 shows UMAP of CD45 cells and there are microglia but I do not see a population activated microglia in addition to homeostatic microglia. I do not see proliferating cells which have been reported in murine glioma models. There is no Tregs population, however in Fig. 3 when T cells are analyzed this population is clearly visible. The authors used signatures from human studies (coming mostly from CYTOF data) to identify macrophage subpopulations,*

instead of signatures used in other scRNAseq mouse studies and it is not clear what Nos2-TAMs mean (ApoE+TAMs likely represent phagocytic cells). None population is marked as immunosuppressive TAMs. Stem-like TILs are enigmatic. Cell to cell interactions analysis (Fig.4d) does not shown the enhancement of specific interactions. The verification Iba1+-TILS interactions is shown on confocal images but it is more co-occurrence than interaction demonstration (Z-stack at high magnification should be demonstrated to prove this point).

Response: Thank you for these comments we will respond below:

1. In response to Referee 1 comment # 3, we reclustered the mononuclear phagocytes to try to resolve additional heterogeneity. In the case of microglia this did not identify any sub-populations within the microglia. We suspect that most of the microglia are activated in response to brain tumor challenge and that any homeostatic microglia were present at such low frequencies that they were not captured in our scRNA-seq data. As a result, the only microglia cluster identified is one that shares characteristics with the disease-associated microglia phenotype, which is an activated microglia phenotype found in various neurodegenerative diseases (Fig. 6c).
2. We regress cell-cycle out during our normalization step, so it is likely that genes associated with proliferation did not strongly impact cell clustering.
3. We suspected that there was heterogeneity within the T cell clusters based on the expression of *Cd4* and *Cd8* in the stem-like TIL cluster, since we wouldn't expect any double-positive cells outside the thymus (Fig. 2d). Indeed, multiple *Cd4* expressing TIL clusters were identified upon reclustering, including regulatory T cells (Fig. 3a). The contribution of the original cell identities to the reclustered identities can be found in Fig. S4a.

4. The gene-signatures we used to annotate cell-types, along with their references are discussed in the scRNA-seq data analysis methods section and provided in supplementary table 7. Briefly Microglia signatures were taken from these two papers, which primarily utilized RNA-seq and scRNA-seq datasets from mice (<https://doi.org/10.1186/s40478-019-0665-y>, <https://doi.org/10.1016/j.celrep.2016.10.052>). A core macrophage gene program signature was taken from this ImmGen paper which leverages murine transcriptomics (<https://doi.org/10.1038/ni.2419>). All other annotation signatures were taken from this paper, which identified conserved cell types found in mouse and human scRNA-seq tumor datasets (<https://doi.org/10.1016/j.immuni.2019.03.009>). All gene signatures were derived from murine transcriptomic datasets.
5. *Nos2*⁺ TAMs were annotated this way due to their high expression of *Nos2* along with canonical macrophage markers. While no TAMs are explicitly marked as immunosuppressive, the *Apoe*⁺ TAMs share characteristics with the Disease-associated microglia phenotype, which is associated with ICB resistance. We additionally demonstrate that several immunostimulatory gene-sets are enriched in the CaMKK2 KO-associated DC-like TAMs relative to the WT-associated *Apoe*⁺ TAMs in Fig. 6b.
6. *Tcf7*⁺ *Cd8*⁺ TILs were discussed as having stem cell-like properties and were found to be associated with ICB response in this seminal paper on the progenitor exhausted to terminally exhausted continuum paradigm (<https://doi.org/10.1016/j.cell.2018.10.038>). The terms stem-like and progenitor exhausted T cells have come to be associated with an ICB-responsive CD8 phenotype (<https://doi.org/10.1038/s41590-022-01219-w>). We try to acknowledge both of these terms in our manuscript.

7. Specific cell-cell interactions can be found in Fig. S3a.
8. Thank you for this suggestion to better demonstrate cell-cell interaction. We have provided a higher magnification image of a single z-plane showing CD4+ and IBA1+ cells interacting in the TME in Fig. S4k.

Major Comment 11: *“In summary, we found that CaMKK2 deficiency dramatically remodels the immune TME to a more anti-tumor, ICB-responsive phenotype, and away from phenotypes associated with ICB resistance”. This general conclusion is disputable as CaMKK2 deficiency (mostly in neurons) may result in smaller tumors and in turn could lead to smaller immunosuppression in the TME allowing stronger ICB response and lesser ICB resistance. I have my doubt if general CaMKK2 KO could provide conclusive answers regarding its effects on the TME. The authors raised this issue in the discussion and attempted to solve it in the experiments in which they show that CaMKK2 deficiency in non-hematopoietic cells (CaMKK2 KO in irradiated recipients) was necessary for any survival benefit. Increased survival could be achieved by combination of CaMKK2 deficiency in both the hematopoietic and non-hematopoietic compartments.*

Response: We appreciate that host CaMKK2 deficiency may reduce tumor size and thus may be indirectly, via immune cell extrinsic factors, altering immune phenotype. We discuss this possibility in response to Referee 1 comment # 2 and #7. In brief, we have provided additional evidence showing that hematopoietic CaMKK2 deficiency is sufficient to produce T-cell phenotypes similar to those found in the context of total CaMKK2 deficiency (Fig. S6b,c). This along with results shown in Fig. 7c suggest that CaMKK2 has pro-tumor roles in the hematopoietic compartment. The primary pro-tumor and ICB-resistance driving role of CaMKK2 seems to be acting through the non-

hematopoietic compartment, and in particular neurons (Fig. 7c-j). The impact of neuronal CaMKK2 on TAM polarization does not seem to be significantly affected by tumor size (Referee 1 comment # 2). Our working model has been expanded upon and included in a primary figure (Fig. 8), in response to Referee 2 comment # 9.

Minor Comment 12: *Please write gene names in italics (i.e. "CaMKK2 expression is associated with a worse prognosis in patients with GBM using The Cancer Genome Atlas (TCGA) and the Chinese Glioma Genome Atlas (CGGA)".*

Response: Thank you for bringing this to our attention. We have italicized all the occurrences of gene names or references to gene expression.

Minor Comment 13: *In the first figure cells are defined as Kluc but in the methods the name is KR158B-Luc, which is slightly confusing and should be unified.*

Response: We appreciate you pointing out this confusing wording. We have made the below revision to clarify.

Indeed, CaMKK2^{-/-} (CaMKK2 KO) mice survived significantly longer than wildtype (WT) mice implanted with three separate GBM tumor models: CT2a, GL261, and KR148B-Luc (Kluc) (Fig. 1d, e, f).

In conclusion, we thank the Referees for their helpful and insightful review of our manuscript. We hope you find the manner in which we addressed the aforementioned concerns is satisfactory for publication in *Nature Communications*.

REVIEWERS' COMMENTS

Reviewer #1 (Remarks to the Author):

The authors sufficiently addressed my concerns

Reviewer #2 (Remarks to the Author):

The authors have provided a detailed consideration of the reviewers' comments. Carefully crafted answers to all questions have been provided concerning the figures, and the text. Many changes have been made to the text and figures thus clarifying the manuscript's contents. The only missing item is to provide a further clarification of the role of CAMKK2 in the bone marrow and how it interacts with the activity of CAMKK2 in neurons. Other than this minor requested change to the proper explanation, I believe the authors have addressed properly all initial queries.

Reviewer #3 (Remarks to the Author):

The authors had addressed properly and in depth all comments of this referee. They provided some additional data to support their interpretation and the correctness of conclusions. I do appreciate their comments and additional analysis done at the request of other reviewers. In most cases I do agree with their comments on positioning of some data/figures in the supplementary instead of the main figures. Regarding the latter, reclustering allowed to distinguish more subpopulations of TAMs (Cluster 0 and 8 are subtypes of DC-like TAMs, and clusters 1 and 2 are subtypes of Apoe+ TAMs) but I wonder if some additional populations i.e. phagocytic or hypoxic TAMs are not appearing. Moreover, the observed differences in TAMs clusters between WT and CaMKK2 KO are interesting and deserve to be presented in main figures.

The manuscript was properly revised and I do appreciate the author's explanations and corrections of some confusing and unclear issues. I believe that in the revised form it is sufficient quality and interest to be published.

Dear Referees,

Thank you for your thoughtful review of our manuscript, which is entitled, “**Neuronal CaMKK2 promotes immunosuppression and checkpoint blockade resistance in Glioblastoma**”. Below, please find a point-by-point response to the italicized comments from the Referees. The attached revision of the manuscript details the changes made in red text and highlights areas in Comment Boxes where we provide specific responses to the concerns of the Referees.

Summary of Figure Changes

Figure 2 – Indications of significance were modified so they could be referred to in the legend.

Figure 3 – Indications of significance were modified so they could be referred to in the legend.

Supplementary Figure 4 - Indications of significance were modified so they could be referred to in the legend.

Figure 6 - Indications of significance were modified so they could be referred to in the legend.

Figure 7 – Indications of significance were modified so they could be referred to in the legend.

Response to Referees

Detailed responses to Referees are outlined below:

Referee 1

General Remarks: *The authors sufficiently addressed my concerns.*

Response: We thank the reviewer for their generosity and thorough review of this manuscript.

Referee 2

General Remarks: *The authors have provided a detailed consideration of the reviewers' comments. Carefully crafted answers to all questions have been provided concerning the figures, and the text. Many changes have been made to the text and figures thus clarifying the manuscript's contents. The only missing item is to provide a further clarification of the role of CAMKK2 in the bone marrow and how it interacts with the activity of CAMKK2 in neurons. Other than this minor requested change to the proper explanation, I believe the authors have addressed properly all initial queries.*

Response: Thank you for your careful review of this manuscript. We have expanded the 2nd paragraph of the discussion section to include the below sentences speculating on how CaMKK2 in neurons may be modulating the immune system. The 2nd to last paragraph of the discussion additionally addresses this topic.

How CaMKK2 in neurons is interacting with the immune system is a topic of great interest. CaMKK2 in neurons likely has direct and indirect immunosuppressive effects. Neuronal CaMKK2 may indirectly influence immunosuppression via supporting tumor growth through the secretion of CaMKK2-dependent mitogenic factors such as BDNF. These larger, faster growing, tumors would be expected to exert stronger immunosuppressive effects. However, when comparing similarly sized tumors, we saw this had no significant effect on the MHCII phenotype of TAMs (data not shown). This indicates that Neuronal CaMKK2's indirect effects are likely not be the primary driver of immunosuppression. Alternatively, neuronal CaMKK2 may be directly immunosuppressive through neuro-immune interactions.

Referee 3

General Remarks: *The authors had addressed properly and in depth all comments of this referee. They provided some additional data to support their interpretation and the correctness of conclusions. I do appreciate their comments and additional analysis done at the request of other reviewers. In most cases I do agree with their comments on positioning of some data/figures in the supplementary instead of the main figures. Regarding the latter, reclustered allowed to distinguish more subpopulations of TAMs (Cluster 0 and 8 are subtypes of DC-like TAMs, and clusters 1 and 2 are subtypes of Apoe+ TAMs) but I wonder if some additional populations i.e. phagocytic or hypoxic TAMs are not appearing. Moreover, the observed differences in TAMs clusters between WT and CaMKK2 KO are interesting and deserve to be presented in main figures.*

The manuscript was properly revised and I do appreciate the author's explanations and corrections of some confusing and unclear issues. I believe that in the revised form it is sufficient quality and interest to be published.

Response: Thank you for your thoughtful review of this manuscript. We are also interested in the different subpopulations of TAMs, and we will keep these comments in mind when designing future studies. We additionally agree that the differences between WT and CaMKK2 KO TAMs are of primary interest. This comparison is the focus of Fig. 6 a-c.

In conclusion, we thank the Referees for their helpful and insightful review of our manuscript. We hope you find the manner in which we addressed the aforementioned concerns is satisfactory for publication in *Nature Communications*.